**Seasonal variability, sources, and parameterization of ice-nucleating particles in the**
**Rocky Mountain region: Importance of soil dust and biological contributions**
**Ruichen Zhou, Russell Perkins, Drew Juergensen, Kevin Barry, Kelton Ayars, Oren Dutton,**
**Paul DeMott, Sonia Kreidenweis**
Department of Atmospheric Science, Colorado State University, Fort Collins, Colorado 80523,
USA
Corresponding author: Russell Perkins (rperkins@colostate.edu)

**Abstract**

Atmospheric ice-nucleating particles (INPs) significantly influence cloud microphysics and aerosol-cloud interactions. Understanding INPs in mountain regions is important for predicting impacts on regional clouds and precipitation. In this study, we conducted comprehensive measurements of immersion-freezing INPs at Mt. Crested Butte in the Rocky Mountains from September 2021 to June 2023 as part of the Surface Atmosphere Integrated Field Laboratory (SAIL) campaign. The average number concentration of INPs active at −20 °C was 2 $L^{-1}$, with distinct seasonal variation characterized by high summer concentrations and low winter concentrations. Aerosol sources were resolved, and INP concentrations were correlated with a coarse dust aerosol type, which dominates $PM_{10}$ in this region. Calculating IN active surface site densities ($n_s$) further supporting the primary contribution from coarse dust to INPs. Treatment with $H_2O_2$ indicated substantial contributions (91% on average) from organic INPs across all activation temperatures, suggesting that supermicron organic-containing soil dust dominates the INPs in this region. Heat-labile INPs, likely biological in origin, were identified as dominant at > −15 °C through heat treatment of samples and showed significantly lower contributions in winter (~96% reduction). Parameterizations based on $n_s$ for the INPs observed in this mountainous region were developed, which effectively reproduced measured INP concentrations, particularly when accounting for seasonal differences. This study provides the first long-term, comprehensive characterization of INPs for the Upper Colorado River Basin region and offers a parameterization potentially useful for predicting INPs in other remote continental regions.

## 1. Introduction

Atmospheric aerosols play critical roles in forming clouds, and inadequate understanding of aerosol-cloud interactions presents one of the largest uncertainties in predicting global climate (IPCC, 2022; Seinfeld et al., 2016). The subset of aerosol particles that serves as ice-nucleating particles (INPs) can trigger heterogeneous freezing of cloud water droplets, allowing them to freeze above the homogeneous freezing temperature (approximately −38°C) (Hoose and Möhler, 2012). Among the mechanisms of heterogeneous freezing, this immersion freezing process is considered the most important process for mixed-phase clouds, which are common at midlatitudes in all seasons (Kanji et al., 2017; De Boer et al., 2011). These INPs are responsible for a significant proportion of initial cloud ice phase formation, thus impacting the Earth's radiative balance and precipitation (Lohmann and Feichter, 2005; Kanji et al., 2017; Burrows et al., 2022). Although INPs are increasingly a focus of study, the sources and abundances of INPs are not well characterized for many regions.

Various aerosol sources have been found to contribute to INPs, including natural and anthropogenic sources (Kanji et al., 2017). Atmospheric mineral dust particles are considered a dominant contributor of INPs throughout much of the troposphere (Murray et al., 2012; Hoose and Möhler, 2012), and they produce high INP concentrations in a mass or surface area basis at temperatures lower than −15°C (Atkinson et al., 2013; Kiselev et al., 2017). Mineral INPs are inorganic components that are lofted from rock or soil and can undergo long range transport to remote areas (Knippertz and Stuut, 2014). In contrast, soil dusts from agricultural or grazed fields (arable soils) were suggested to contribute 25% of the global dust burden (Ginoux et al., 2012), and were found to initiate ice nucleation at temperatures as high as −6°C (Garcia et al., 2012; O'Sullivan et al., 2014). Organics in arable soil dust are suggested as the main contributors to their

ice nucleating ability (Tobo et al., 2014; Hill et al., 2016). Biomass burning aerosols and fly ash
also contribute to INP populations and have received increasing attention under global warming
and the accompanying more frequent and intense wildfires (Prenni et al., 2012; McCluskey et al.,
2014; Umo et al., 2015). Biomass burning aerosols typically present lower ice nucleating ability,
defined as the IN active surface site density for ice nucleation on particle surfaces (i.e., INP
concentration/aerosol surface area concentration), compared to dust particles, while atmospheric
aging can potentially enhance their ice nucleating ability (Jahl et al., 2021). Biogenic aerosols,
such as primary biological particles composed of bacteria, pollen, fungal spores, and their
fragments, were identified in the 1970s as important INP sources (Vali and Schnell, 2024; Schnell
and Vali, 2024). They typically can activate ice formation at a warmer temperature than other INPs
listed above and thus may control first ice formation in clouds (Pratt et al., 2009; Creamean et al.,
2013; Tobo et al., 2013). Besides these INP sources, marine aerosol (Wilson et al., 2015;
McCluskey et al., 2018b), secondary organic aerosol, and fuel-combustion aerosols can contribute
to INPs (Kanji et al., 2017).
To estimate INPs for use in numerical cloud models, early parameterizations were typically based
on empirical relationships between INPs and temperature or supersaturation alone (Bigg, 1953;
Meyers et al., 1992). These parameterizations could show large biases compared to field
observations. DeMott et al. (2010) proposed a widely used parameterization based on temperature
and number concentrations of particles larger than 0.5 μm in diameter, which has been applied in
global and regional models due to its convenience and independence from detailed aerosol
composition (Miltenberger et al., 2018; Storelvmo et al., 2011; Burrows et al., 2022). However,
recent studies highlight the need for more physically based and source-specific parameterizations
(Burrows et al., 2022; Shi and Liu, 2019; DeMott et al., 2015). Laboratory and field studies have
shown that different aerosol types (e.g., mineral dust, biological particles, marine aerosols) exhibit
distinct ice-nucleating efficiencies (Hoose and Möhler, 2012; Kanji et al., 2017). Parameterizations
based on the ice-nucleating IN active surface site density ($n_s(T)$) have been developed for various
aerosol types under immersion freezing conditions (Niemand et al., 2012; DeMott et al., 2015;
Harrison et al., 2019; McCluskey et al., 2018a; Umo et al., 2015; Schill et al., 2020; Tobo et al.,
2014; O'Sullivan et al., 2014). Further field observations and laboratory studies are needed to
improve these parameterizations and reduce uncertainties in INP predictions.
Mountainous regions, covering approximately one-quarter of the global land surface, play a critical
role in regional and global hydrological and climatic systems. Importantly, they are sources of
freshwater supporting nearly half of the world's population (Viviroli et al., 2007). Cloud and
precipitation formation in mountainous areas are strongly influenced by aerosol–cloud interactions
(Lynn et al., 2007). The presence, variability, and sources of INPs in mountain areas can
significantly affect cloud phase, lifetime, and precipitation efficiency (Creamean et al., 2013; Lynn
et al., 2007). However, INP observations in mountain environments, especially for long-term
continuous measurements, are limited (Lacher et al., 2018; Sun et al., 2024; Conen et al., 2015),
hampering our understanding of INP major sources, seasonal variation, and the influence of
complex mountain terrain on their vertical distribution. Improved understanding of INPs in
mountainous regions is essential for better representation of clouds and precipitation formation in
weather and climate models, and for predicting future changes in water availability under global
climate change regimes.
The Surface Atmosphere Integrated Field Laboratory (SAIL) Campaign, conducted from
September 2021 to June 2023 in the Upper Colorado River Basin of the Rocky Mountains,
included aims to improve understanding of how aerosols, particularly long-range transported dust
and wildfire smoke, affect the surface energy and water balance through their impacts on cloud,
precipitation, and surface albedo, and how these effects vary by season (Feldman et al., 2023).
This campaign provided a unique opportunity to investigate INPs in the Colorado Rocky
Mountains over a nearly two-year period. This study presents comprehensive measurements of
immersion-freezing INPs, including their seasonal and temperature-dependent variability, as well
as associations with aerosol sources. We further explore different INP compositional types,
biological/heat-labile, other organic, and inorganic INPs, and their inter-relationships, highlighting
the importance of organic INPs. A parametrization method is proposed for INPs in this region that
reproduces the observed two-year INP concentration record.

## 111 2. Methods

### 112 2.1 Sampling site and sample collection

The SAIL campaign deployed the Department of Energy Atmospheric Radiation Measurement
(DOE ARM) Mobile Facility 2 (AMF-2) at the East River Watershed, which is located near Crested
Butte and Gothic, Colorado (Feldman et al., 2023). This mountainous region, with elevations
ranging from ~2440 to 4350 m above sea level, is characterized by complex terrain, a deep seasonal
snowpack, and pronounced hydrometeorological gradients. The region experiences strong
seasonal contrasts, with cold snowy winter and warm summers influenced by convective activity
associated with the North American monsoon (Feldman et al., 2023).
For INP analyses, aerosol filter samples were collected by DOE ARM technicians through the INP
Mentor Program approximately every three days from September 2021 to June 2023, with each
sampling period lasting about 24 hours, as described in the instrument handbook (Creamean et al.,
2024) and repeated briefly here. Aerosols were collected on two 47 mm Nuclepore polycarbonate
filters (0.2 μm pore size) at flow rates in the range of 10–18 lpm, with total sampling volumes of
approximately 15000 to 25000 L. For computing atmospheric concentrations in this study, the
sampling volume was corrected to standard temperature (273.15 K) and pressure (101.3 kPa). Prior
to sampling, the filters were cleaned using 10% $H_2O_2$ and deionized (DI) water to remove organic
and biological residues, then stored in sterile Petri dishes. For the first month (September 2021) of
the campaign, samples were collected at the M1 site (38°57′22.35″N, 106°59′16.66″W; 2885 m
above mean sea level (MSL)). From October 2021 to June 2023, samples were collected at the S2
site (38°53′52.66″N, 106°56′35.21″W; 3137 m MSL). The distance between M1 and S2 sites is
about 8 km. The possible difference in INP concentrations at M1 in the first month is not addressed
here, since this study is aimed at representing the INPs in this region, and there was no significant
change in INP concentrations when moving from the M1 to the S2 site. All samples were stored at
−20 °C after collection, during shipment, and until the analysis in the laboratory.

## 2.2 Laboratory Ice Spectrometer analysis of filter-collected particles

The immersion freezing ability of particles collected on filters was quantified using the Colorado
State University (CSU) Ice Nucleation Spectrometer (INS), following established procedures
(McCluskey et al., 2017; Hiranuma et al., 2015; Barry et al., 2021a), which are the same methods
used by the DOE ARM INP Mentor Program (Creamean et al., 2024). The INP Mentor Program
analyzed many of the samples from the SAIL campaign. Our analysis was used to fill gaps in time
for the Mentor Program samples, and provide additional heat and hydrogen peroxide treatments,
described below. Our data on these samples is provided within the Mentor Program data product
(Shi et al., 2025). Briefly, exposed filters were mixed with 10 mL of filtered deionized water in a
centrifuge tube, then rotated for 20 minutes using an end-over-end shaker to resuspend the particles.
Aliquots of the resuspension solution were pipetted into a 96-well PCR tray and cooled at a rate
of 0.33 °C min$^{-1}$ from room temperature to −30 °C in the CSU INS. The number of frozen wells
was recorded at 0.5 °C intervals. Cumulative INP concentrations as a function of temperature
($n_{INPs}(T)$, INPs per liter of air) were calculated based on the method of Vali (1971) using:
$$n_{INPs}(T) = ln(\frac{N_0}{N_0 - N(T)}) \times \frac{V_w}{V_c} \times \frac{1}{V_a}$$

where $N_0$ is the total number of wells containing aliquots, $N(T)$ is the cumulative number of wells
frozen at temperature $T$, $V_w$ is the volume of water used for particle resuspension, $V_c$ is the aliquot
volume added to each well, and $V_a$ is the total sampled air volume. Counting uncertainties were
estimated using binomial confidence intervals (Agresti and Coull, 1998). Field blank filters were
collected every month by briefly exposing them at the sampling site for several seconds before
storage. Before calculating INP concentrations, the average number of INPs versus temperature
per blank filters was subtracted from the calculated number of INPs versus temperature per sample
filter, to account for potential contamination during sampling and handling, as well as any residual
contamination on the filters after cleaning.
To further characterize the types of INPs, portions of the suspended aerosol solution were subjected
to heat and hydrogen peroxide ($H_2O_2$) treatments, followed by freezing analysis of aliquots of these
portions. For heat treatment, the solution was heated to 95 °C for 21 min before measurement. This
process inactivates the biological INPs by denaturation of proteins and removes heat-labile INPs
(O'Sullivan et al., 2014; Tobo et al., 2014; Hill et al., 2016). In total, 43 samples were exposed to
95 °C heat treatment, and their temperature spectra are shown in Figure S1. In the peroxide
treatment, 30% $H_2O_2$ was added to the solution to make a final concentration of 10%, and the
mixture was heated at 95 °C for 21 min under UVB light to digest organics (Suski et al., 2018),
and the INPs remaining were presumed to be inorganic. A total of 34 samples underwent $H_2O_2$
treatment, with their temperature spectra shown in Figure S2. By comparing untreated (base), heat-
treated, and $H_2O_2$-treated results, INPs were categorized into biological/heat-labile, other organic,
and inorganic types. Daily et al. (2022) found that some minerals also showed reduced immersion-
freezing activity after heat treatment; however, SAIL samples showed some difference with the
behaviors they reported. For minerals with initial active temperatures > −10 °C, IN active surface
site density either decreased at all measured temperatures (Arizona Test Dust (ATD) and Fluka
Quartz) or was not sensitive to wet heating (K-feldspar) (Daily et al., 2022), differing from some
spectra of SAIL samples that only showed decreases at warm temperatures and almost no change
at temperatures < −18 °C (Figure S1). This suggests that mineral INPs have limited contributions
to the decreases of INP concentrations after heat treatment in the SAIL samples.
There were eight ground-based sites within 55 km of the SAIL sampling locations conducted
orographic cloud seeding operations by North American Weather Consultants Inc., targeting
precipitation enhancement. To the best of our knowledge, these seeding stations combusted
solutions in a propane flame, producing particles containing silver iodide (AgI) and other inorganic
salts that served as seeding aerosols. These seeding activities occurred in specific storm situations
during winter and early spring and strongly impacted our INP observations. The measured
concentrations of INPs active at temperatures from −7.5 °C to −27.5 °C for a total of 113 24-hour
samples are shown in Figure S3. Since this work focuses on investigating the natural or background
INPs in the Rocky Mountain region, samples collected on days that overlapped with artificial cloud
seeding activities, as recorded in their logbook (data provided by North American Weather
Consultants Inc.), are highlighted in Figure S3 and excluded from the discussion below. Cloud
seeding activities last less than 24 hours, typically 4-8 hours, and are unlikely to affect the
subsequent sample collected 3 days later. Also, eight samples collected during winter exhibited
distinct INPs spectra from other samples, but highly similar to the INP spectrum of Snomax®, a
commercial non-living bacterial INP product used in snowmaking (Figure S4a). Furthermore, *P.*
*syringae,* the bacterium type in Snomax®, was identified in some of these samples based on qPCR
analysis (Supplement Text S2, Figure S4b). These samples are highly suspected to have been
affected by snowmaking activities during wintertime, associated with the location of sampling site
within a ski resort at Crested Butte. Therefore, these samples, along with those affected by cloud
seeding activities, were excluded from the subsequent discussion to better understand the
characteristics of natural INPs in the Rocky Mountain region.

**2.3 Source apportionment**
To investigate aerosol sources in the SAIL region, source apportionment was performed using
positive matrix factorization (PMF). PMF is a receptor model that decomposes an observation
matrix into factor profiles and their corresponding contributions related to emission sources and/or
atmospheric processes, providing a quantitative assessment of source influences (Paatero and
Tapper, 1994). Data from the Interagency Monitoring of Protected Visual Environments
(IMPROVE) site at White River, located approximately 30 km north of the SAIL campaign site,
were used for PMF analysis. The IMPROVE network (Malm et al., 1994) has collected 24-hour
aerosol filter samples every three days over several decades at this site, providing a valuable dataset
to understand aerosol sources and their long-term variability in the region. Elemental analysis was
performed on the Teflon filters using X-ray fluorescence (XRF), anions were analyzed using ion
chromatography (IC), elemental carbon (EC) and organic carbon (OC) were analyzed using a
carbon analyzer (Hand, 2023). Chemical concentrations in the $PM_{2.5}$ fraction of nineteen elements
(Al, As, Br, Ca, Cl, Cr, Cu, Fe, K, Mg, Mn, Na, Ni, Pb, Se, Si, Ti, V, and Zn), along with nitrate,
sulfate, elemental carbon (EC), organic carbon (OC), and calculated coarse mass concentrations
($PM_{10}-PM_{2.5}$ mass concentrations), from January 2014 to April 2024 were used as input for the
PMF analysis performed using EPA PMF 5.0 (Norris et al., 2014). IMPROVE species
concentrations were reported based on local conditions.
A five-factor solution was selected as the optimal solution based on the $Q/Q_{exp}$ value and
interpretation of the physical meanings of the factors (Brown et al., 2015). The corresponding
factor profiles and time series are shown in Figures S5 and S6. These factors were identified, based
on chemical signatures and previous literature, as coarse dust, fine dust, biomass burning, sulfate-
dominated, and nitrate-dominated sources. Coarse and fine dusts had high contributions from Al,
Ca, Fe, Mg, and Si, which are the main components of mineral dust (Liu and Hopke, 2003). Coarse
dust explained more than 90% of the coarse mass ($> PM_{2.5}$), while there was no contribution from
coarse mass in the fine dust factor. The biomass burning factor was strongly associated with
organic and elemental carbon, which are mainly from combustion processes, and K, a tracer of
biomass burning (Hopke et al., 2020). The other two factors are dominated by nitrate and sulfate,
which are related to the formation of secondary aerosols and possibly some primary emissions
from regional sources that include energy production and distant urban regions. Some similar
factors were also resolved in published PMF analyses using IMPROVE data (Liu and Hopke, 2003;
Hwang and Hopke, 2007). Further details on the PMF analysis and results, as well as support for
their applicability over the broad surrounding Rocky Mountain region (IMPROVE sites at Mount
Zirkel and Rocky Mountain National Park) are provided in Supplement Text S1 and Figure S7.
Data for the PMF results are available through the ARM data product (Zhou et al., 2025a).
To assess the impact of different sources on INPs, the INP samples were categorized into six types
based on the dominant aerosol sources during the sampling period, and the contribution of sources
to each sample was shown in Figure S8. However, IMPROVE samples were taken only one of
every three days. For days without available IMPROVE data, aerosol sources were inferred from
the nearest sampling days. Samples were classified as follows: (1) Coarse dust, if coarse dust
contributed more than 50% to the total PM10 mass concentration or contributed more than 40%
and represented the largest contribution among all sources; (2) Biomass burning, if biomass
burning accounted for more than 50% of the total $PM_{10}$; (3) Dust, if the combined contribution of
coarse and fine dust exceeded 50% of the total $PM_{10}$, with the difference between fine and coarse
dust being less than 20%; (4) Fine dust, if the combined mass contribution of coarse and fine dust
exceeded 50% of the $PM_{10}$ and fine dust exceeded coarse dust by more than 20%; (5) Mixed
samples, samples for which no single source was dominant, typically characterized by a sulfate
contribution greater than 20% of the $PM_{10}$. (6) Additionally, seven samples fell within periods
where more than one consecutive IMPROVE sample was missing (one week or longer). For these
cases, source contributions were not interpolated, and they were categorized as samples with no
source data. Source influences inferred from the nearest sampling days may introduce uncertainties.
However, the merged size distribution data (section 2.5 and Figure 2) for these days showed similar
size distribution patterns and comparable number concentrations for most size ranges compared to
the nearest days, suggesting that significant changes in aerosol sources for these inferred days were
limited. Also, the discussion is based on each group containing multiple samples, which should
also reduce the uncertainties associated with the inferred source from a single sample.

**2.4 Back trajectory analysis**
Air mass back trajectory analysis was performed using the Hybrid Single-Particle Lagrangian
Integrated Trajectory model (HYSPLIT; Stein et al., 2015; Rolph et al., 2017), developed by the
National Oceanic and Atmospheric Administration (NOAA) Air Resources Laboratory. For each
hour during each sampling period (typically 24 h), a 96-hour back trajectory was initiated at the
SAIL sampling site, starting 100 m above ground level, and using the GDAS meteorological
dataset with model vertical velocity. The areas traversed by the back trajectory were gridded into
$1° \times 1°$ cells. For each trajectory, its occurrence in each grid cell was weighted by the residence
time spent in that cell. To account for the peak in occurrence near the sampling site, the residence
times were further normalized by the distance from the SAIL sampling site, following the method
of Ashbaugh et al. (1985). The analysis was performed for each sample, and the resulting
trajectories were aggregated to produce a composite residence-time map (Figure 1). Separate
analyses were also performed for samples categorized by different source types (Figure S9),
aggregating hourly trajectories for all sample times of a corresponding type across the campaign.
Back trajectory data are available through the ARM data product (Zhou et al., 2025b).

**2.5 Merged particle number-size distribution and IN active surface site density**
During the SAIL campaign, a scanning-mobility particle sizer (SMPS) and an optical particle
counter (OPC) were deployed simultaneously with the filter sample collection to measure aerosol
number size distributions in the particle diameter ranges from 10–500 nm and 0.25–35 μm,
respectively (Kuang et al., 2024; Cromwell et al., 2024). To obtain a continuous number size
distribution from 10 nm to tens of micrometers, measured number size distributions from the
SMPS and OPC were merged following previous methods (Hand and Kreidenweis, 2002;
Marinescu et al., 2019). Briefly, the mobility diameters from the SMPS were assumed to be equal
to volume equivalent diameters by assuming the particles are spherical, and number distributions
were converted to volume distributions. A scaling factor was determined by comparing the
overlapping size range of the two instruments, and the OPC volume size distribution was then
aligned with the SMPS volume distribution measurements by shifting OPC measured diameters to
estimate the SMPS mobility diameter corresponding to that optical diameter. All aerosol data from
the ARM archive were corrected to standard temperature (273.15 K) and pressure (101.3 kPa). All
merged size distribution data are available through the ARM data product (Zhou et al., 2025c). A
timeline of the merged number-size distributions of aerosols during the SAIL campaign is shown
in Figure 2.
Assuming that the number of active ice nucleation sites is linearly proportional to the particle
surface area, the IN active surface site density (m$^{-2}$) at temperature $T$ ($n_s$(T), m$^{-2}$) was calculated
using the following equation:
$$n_{s,500}(T) = \frac{n_{\mathrm{INP}}(T)}{S_{\mathrm{m},500}} \times 10^9$$

where $n_{\mathrm{INP}}$(T) (sL$^{-1}$) is the measured INP concentration at temperature $T$, $S_{\mathrm{m},500}$ (μm$^2$/scm$^3$) is the
surface area concentration of particles larger than 500 nm diameter calculated from the merged
size distribution, assuming that particles are spherical, and the $10^9$ conversion factor is used for
$n_{s,500}$ units of m$^{-2}$. In this study, the surface area of particles > 500 nm is used to exclude surface
area associated with pollution and other aerosol types that are inefficient sources of INPs. This is
supported by the correlation found between INP concentrations and the number concentrations of
particles > 500 nm in a previous study (DeMott et al., 2010). The $n_s$ is also calculated based on
total surface area concentrations to facilitate comparison with other studies and is shown in Figure
S10. While the IN active surface site density approach is typically fully justified for single INP
compositions, we will apply it here to the total surface areas, but also discuss adjustments needed
when comparing to more specific INP parameterizations.

## 3. Results and Discussion

### 3.1. INPs concentrations at the SAIL study site

Samples not affected by artificial INP generation activities (cloud seeding and snowmaking
activities in winter; see section 2.2), representing the natural INPs in this region (84 samples in
total), are shown in Figure 3a. The discussion in this study focuses on these samples to better
understand the characteristics of natural INPs in the Rocky Mountain region. INP concentrations
ranged from $4 \times 10^{-4}$ L$^{-1}$ to 1.5 L$^{-1}$ (mean: 0.15 L$^{-1}$, median: 0.05 L$^{-1}$) at $-15$ °C, and from 1.2
L$^{-1}$ to 90 L$^{-1}$ (mean: 16 L$^{-1}$, median: 12 L$^{-1}$) at $-25$ °C. This is comparable to online INP
measurements in the Rocky Mountain region (median: 8.2 L$^{-1}$ at $-26$ °C; Lacher et al., 2025).
Compared with previous INP studies summarized by Kanji et al. (2017), the INP concentrations
observed during the SAIL campaign fall within the range of those in their compilation that were
influenced by dust, biomass burning, and precipitation, and are higher than those from marine
aerosols but lower than the maximum of those attributed to biological sources. Regarding temporal
variations, INP concentrations at all measured temperatures followed a similar trend over the
nearly two-year observation period: low in winter, increasing in spring, and reaching highest
concentrations during summer and early fall. For activation temperatures warmer than $-25$ °C, the
highest INP concentrations were all observed in summer. For temperatures colder than $-25$ °C, the
samples from summer also showed high INP concentrations, while the peak concentrations
occurred in September 2021. Back trajectories of those samples (2021-9-9 and 2021-9-16) showed
that the air masses mainly originated from the northwestern U.S. Intense wildfires occurred in that
region during the summer of 2021 (Jain et al., 2024), and the transported smoke plumes increased
aerosol loading at the SAIL site. These smoke intrusions may also result in enhanced INP
concentrations active at low temperatures.
Monthly mean INP concentrations were calculated and are presented in Figure 3b, to better
visualize seasonal trends and reduce the impact of individual outliers. INP concentrations showed
clear seasonal variations throughout the campaign. Note that our sampling did not cover the entire
month, so some outliers may have been coincidently captured or missed, however, the two-year
monthly dataset is still expected to broadly represent the seasonal variations of INPs in this region.
In general, INP concentrations reached peaks in June 2022 across all temperatures, and reached
minimum levels during winters in both years. However, differences existed among activation
temperatures. At warmer temperatures (−10 °C and −15 °C), INP concentrations were relatively
similar in warm seasons (April–October) and much higher than those in cold seasons (November–
March). Bioaerosols are typically recognized as major INP sources at these activation temperatures
(Kanji et al., 2017), and their emissions generally decrease in winter in most areas due to reduced
biological activities and possible snow cover limiting resuspension (Fröhlich-Nowoisky et al.,
2016). A distinct high INP peak was observed in June 2022, which was ten times higher than that
in June 2023, and much higher than in other months, suggesting that a high INP emission event
occurred during this month, which may be related to a specific biological emission event and/or
dust event. For activation temperatures of −20 °C and −25 °C, INP concentrations also showed a
seasonal pattern increasing from April, peaking in June, then decreasing and reaching a minimum
in December. Recent online INP measurements for activation temperatures from −22 °C to −32 °C
conducted from October 2021 to May 2022 and January to May 2025 at the Storm Peak Laboratory
in the Rocky Mountains (Lacher et al., 2025) found a similar seasonal pattern, with the lowest INP
concentrations in winter and increased in spring, suggesting that the INP sources could be similar
and may dominate INPs across a broaden region of the Rocky Mountains. The elevated INPs from
April to September may be attributed to enhanced dust aerosols, as dust concentrations were found
to increase during this period (Hand et al., 2017), and dust is a significant source of INPs,
especially at temperatures below −15 °C (Kanji et al., 2017; DeMott et al., 2015). Lower-
temperature INP concentrations in June 2022 were also higher than that those in June 2023, while
the magnitude of this difference (a factor of two) was less distinct from the difference at −10 °C
(twelve times). This suggests that the origins of INPs activated at lower temperatures differ from
those at warm temperatures.

**3.2. Relationships between aerosol sources and INPs**
The sources of aerosols to the SAIL campaign region were identified based on the PMF analysis
(Text S1, Figures S5 and S6). Different sources exhibited unique seasonal trends. Coarse dust
showed increased concentrations from April to September and had the highest annual mean
concentration among the five resolved particle types. Fine dust increased sharply in April, May,
and June, and remained low during other months. Biomass burning aerosol varied significantly by
year. Strong peaks in concentrations were observed in September 2021, with SAIL collecting some
samples at the end of these events, and increased contributions in June–August 2022. The biomass
burning factor typically showed high concentrations in summer. The sulfate-dominated and nitrate-
dominated factors had much lower concentrations overall, with peaks occurring in summer and
spring, respectively. The aerosol number size distribution also reflected the variations in different
sources (Figure 2). Supermicron particle concentrations were higher when coarse dust increased.
Submicron particles concentrations increased in September 2021 and the summer of 2022, which
corresponded to increases in contributions from the biomass burning factor. To investigate the
impact of different aerosol types on INPs, temporal variations between aerosol sources and INP
concentrations at −15 °C and −25 °C were compared, as shown in Figure 4. Monthly average
concentrations were used to minimize the impact of unsampled dates. Although this may introduce
uncertainties by smoothing episodic peaks of a source, the nearly two-year (22 months) record
should adequately represent its seasonal cycle. Pearson correlation coefficients between monthly
INP concentrations (−10 °C, −15 °C, −20 °C, and −25 °C) and aerosol sources are shown in Table

381  1.

Coarse dust and biomass burning presented seasonal variations similar to those of INPs at −15 °C
and −25 °C, while fine dust, nitrate-dominated, and sulfate-dominated factors had weaker
correlations (Figure 4). The strong correlations with coarse dust and biomass burning aerosols
suggest that these sources significantly contributed to observed INPs. Coarse dust showed good
correlations with INPs active at all temperatures (Table 1), with correlation coefficients increasing
for colder temperatures, suggesting that coarse dust is a major source of INPs, particularly at lower
temperatures. This is consistent with previous findings that dust dominates the INPs at
temperatures below −20 °C (Beall et al., 2022; Testa et al., 2021; Kanji et al., 2017). Furthermore,
Lacher et al. (2025) provided direct evidence that INPs active at cold temperatures were
significantly contributed by supermicrometer particles, which they attributed to dust, in the Rocky
Mountains. Their observation site was located near to the IMPROVE site at Mount Zirkel, where
our PMF analyses identified similar sources and trends to those near the SAIL (Text S1),
suggesting that INPs in both studies were impacted by coarse dust. Interestingly, coarse dust
presented a weaker correlation with INPs at −10 °C ($R^2$ = 0.43), a temperature range usually
associated with biological INPs. This may be due to the large number of coarse dust particles,
biological INPs carried on dust particles, and/or the inclusion of biological particles in the coarse
dust factor, as biological particles are mostly supermicron in size (Després et al., 2012). Biomass
burning presented a strong correlation with INP at −25 °C and weak correlations at warmer
temperatures (−10 °C and −15 °C), suggesting that aerosols from biomass burning contribute
primarily to INPs active at lower temperatures. Combining coarse dust and biomass burning
contributions showed an even better correlation with INPs (Table 1), supporting that these are the
major contributors of INPs in this region, especially at lower activation temperatures.
Different from coarse dust, fine dust showed weak relationships with INPs at all temperatures ($R^2$
= 0.10–0.19), suggesting lower contributions to INPs. One possible reason is that larger particles
have more ice nucleation active sites (Reicher et al., 2019; DeMott et al., 2010). Another reason
could be differences in sources of fine and coarse dust that resulted in different ice-nucleating
abilities. Fine dust presented a different seasonal pattern compared to coarse dust. Fine dust
concentrations peaked in spring, especially in April, while coarse dust was higher in summer. This
difference was also observed in Hand et al. (2017) for the Colorado Plateau and Central Rockies
regions. They suggested that fine dust in this region is influenced by regional or long-range
transported dust, such as Asian dust, while coarse aerosol mass concentrations (defined as the
difference between $PM_{10}$ and $PM_{2.5}$, which was almost all loaded into the coarse dust factor in our
PMF analysis) are mainly derived from local and regional sources. Sulfate-dominated and nitrate-
dominated sources were not correlated with INP concentrations. In remote areas, sulfate and nitrate
particles mainly come from secondary formation (Seinfeld and Pandis, 2016) and are generally
not considered as efficient INPs (Kanji et al., 2017).

**3.3. INP temperature spectra categorized by sources**

The total INP temperature spectra categorized by dominant aerosol types are shown in Figure 5a, and total INP spectra sorted under dominant influence of each source type are plotted separately in Figure S11. Significant differences in INP concentrations were observed between periods where different aerosol source types were dominant. Over 50% of the samples were dominated by coarse dust, which was the predominant aerosol source in this region (Figure S8). Considering that coarse dust also showed a strong correlation with INPs, this suggests INP concentrations were likely primarily influenced by coarse dust in this area. Mineral dust (DeMott et al., 2003; Niemand et al., 2012; Atkinson et al., 2013), soil dust that contains abundant organics (Tobo et al., 2014; Steinke et al., 2016; O'Sullivan et al., 2014), and playa dusts (Pratt et al., 2010; Hamzehpour et al., 2022) have been widely investigated, and are considered as important INP sources. Based on back trajectories for samples categorized as coarse dust (Figure S9), air masses were mostly from local or regional sources (Central Rockies and Colorado Plateau, with additional inputs from the agricultural Imperial Valley in southern California). Considering their larger size and mass, coarse dust particles have relatively short atmospheric transport ranges, and local resuspension of soil is likely a dominant source (Hand et al., 2017). High air mass residence times were also indicated for the Great Salt Lake region, suggesting potential contributions from playa salts. This is consistent with the contributions of Cl and Na in the coarse dust factor profile from the PMF analysis (Figure S5a), and playa salt dusts have also been observed, at least at lower temperatures in the mixed phase regime, to serve as INPs (Pratt et al., 2010; Koehler et al., 2007).

Compared to INPs samples that aerosols dominated by coarse dust, fine dust-dominated time periods showed lower INP concentrations (Figure 5). Back trajectories also indicated different origin regions from those for coarse dust; samples dominated by fine dust had trajectories mainly

from southern California and the Sonoran Desert in Arizona, known as an area with high dust
emissions (Ginoux et al., 2012). Fine dust is likely long-range transported as Asian dust (Hand et
al., 2017) or from the noted southwestern desert areas. Besides their different origins, the higher
INP concentrations associated with coarse dust-dominated samples compared to those dominated
by fine dust can also be attributed to differences in particle size, although confirming this would
require data on size-resolved INPs that were not collected. INPs related to biomass burning-
dominated samples presented comparably high concentrations, which may be related to the
significantly elevated aerosol loading during biomass burning events. To assess the ice nucleating
activity, with the influence of aerosol concentrations and size distributions accounted for, the IN
active surface site density was further investigated.

**3.4. IN active surface site density ($n_s$) temperature spectra**
The IN active surface site density ($n_s$) temperature spectra, a measure of INPs per aerosol surface
area, are shown in Figure 5b for all samples, categorized by dominant aerosol type. Compared to
INP concentrations, the $n_s$ values were less variable at a given temperature, with most samples
within one order of magnitude of each other, whereas INP concentrations spanned nearly two
orders of magnitude, suggesting that at least some of the variability in INP concentrations can be
explained by differences in particle size distributions and concentrations.
Samples dominated by a specific aerosol source (coarse dust, dust, fine dust, or biomass burning)
exhibited relatively consistent $n_s$ values with some differences among categories. The $n_s$ of samples
dominated by coarse dust was similar to or slightly higher than those having both abundant coarse
and fine dusts (categorized as dust), suggesting that INPs from coarse dust have higher IN active
surface site density. After normalizing by surface area, the $n_s$ for the fine dust-dominated samples
showed closer values with those of the coarse dust-dominated samples, while still lower. This
suggests that the lower INP concentrations in fine dust-dominated samples can be partly attributed
to differences in aerosol surface area concentrations, but also to lower active site density due to
potentially different INP sources. The differences in $n_s$ between coarse dust- and fine dust-
dominated samples were limited, likely because there were still small contributions from coarse
dust (17% on average), although fine dust dominated these samples (59% on average).
During biomass burning events, aerosol number concentrations were significantly enhanced,
especially for submicron particles (Figure 2). A correlation was found between the biomass
burning factor mass concentrations and the total surface area concentrations of aerosols (Figure
S12), suggesting that such events significantly increased aerosol surface area concentrations.
However, its contributions to INPs could be affected by coarse dust. After normalization by surface
area ($S_{m,500}$), the $n_s$ of INP samples for aerosols dominated by biomass burning were similar to
those of coarse dust-dominated samples. Compared with previous studies (comparison based on
computing $n_s$ using total surface area), these values were higher than $n_s$ reported from laboratory
biomass burning studies (Umo et al., 2015; Jahn et al., 2020) and those reported in ambient biomass
burning observations (McCluskey et al., 2014; Barry et al., 2021b; Zhao et al., 2024). During SAIL,
this finding may have been due to the presence of coarse dust, which has a much higher $n_s$ than
biomass burning, as this aerosol type still contributed moderately to the total aerosols in these
samples (an average of 22% of $PM_{10}$), although biomass burning was the dominant aerosol source
(62% on average). Wildfire events could also be a source of airborne dust (Wagner et al., 2018;
Meng et al., 2025). From the NOAA Hazard Mapping System (Figure S13) and back trajectories
(Figure S9), these samples were mostly affected by long-range transported biomass burning
aerosols originating primarily from wildfires in the northwest and southwest U.S. Aging could
enhance the $n_s$ of biomass burning INPs (Jahl et al., 2021) and may also have contributed to higher
$n_s$ of these samples.
In contrast to the convergence observed in samples dominated by a single source, samples related
to mixed sources showed no reduction in variability after normalization by surface area (Figure 5
and Figure S11). Although it is difficult to precisely determine the INP sources for these samples
based on current available analyses, this comparison strongly supports the link between INPs and
the dominant bulk aerosol sources in this study.

**3.5 Evidence of biological contribution to INPs**
From the INP temperature spectra and $n_s$ for all samples (Figure 5), the spectra for INPs active at
temperatures higher than about −18 °C showed a segregation into two groups: one with higher INP
concentrations (and $n_s$) and measured detectable freezing onset temperatures mostly > −10 °C, and
the other with lower INP concentrations (and $n_s$) and lower measured detectable freezing onset
temperatures, with most of those samples assigned to aerosol sources of mixed and fine dust. We
found all samples in the latter group were collected from December to March. At freezing
temperatures warmer than −15 °C, biological INPs are likely to play a more important role (Kanji
et al., 2017).
Here, we separated samples collected during cold seasons (December–March) and during other
seasons (April–November), as shown in Figure 6. This separation clearly shows that on average,
INP concentrations were lower in cold seasons, with the most striking difference at temperatures
warmer than −15 °C (15% of INPs in other seasons). A further comparison of $n_s$ (Figure 6b) showed
that samples from cold seasons had similar $n_s$ at temperatures colder than around −18 °C. However,
at warmer temperatures, samples from cold seasons showed consistently lower $n_s$. These results
suggest that INPs that activated at temperatures colder than around −18 °C likely originated from
similar sources throughout the whole year, which were primarily associated with coarse dust, as
discussed above. However, in cold seasons, the contribution from biological INPs was
significantly reduced, leading to the divergence in the spectra for temperatures warmer than around
−18 ˚C. This seasonal pattern is supported by the environmental temperature dependence of
biological aerosol emissions (Shawon et al., 2025).
The likely biological nature of these warmer-temperature INPs was also identified from the heat
treatment of samples. A compilation of spectra from all base (untreated) analyses, heat, and $H_2O_2$
treatments is shown in Figure 7. The difference between the base and heat spectra indicated a large
contribution (82–94%, 90% on average) of heat-labile INPs at warm temperatures (> −15 °C),
which are presumably biological INPs. Each individual heat spectrum is shown in Figure S1. In
September 2021, most samples showed decreased INP concentrations at temperature warmer than
−15 °C after heat treatment, while almost no decrease was observed at lower temperatures. This
reduction is likely due to denaturing of biological INPs, reducing or removing their IN activity
(Hill et al., 2016). Samples from September 2021 were strongly affected by biomass burning
(Figure S6). Kobziar et al. (2024) found that biological aerosols can be co-emitted in biomass
burning. The presence of biological INPs in smoke plumes was also suggested in aircraft
measurements of biomass burning aerosols, which showed base and heat-treated spectra (Barry et
al., 2021b) similar to those observed in SAIL samples from September and October 2022. During
the SAIL campaign, Shawon et al. (2025) and Ashfiqur et al. (unpublished data) detected biological
aerosols using a Wideband Integrated Bioaerosol Sensor (WIBS) and scanning electron

microscopy during selected intensive observation periods (June–September 2022 and September 2022, respectively), further supporting the presence of biological aerosols during these periods, some of which may be active as INPs. Biological/heat-labile INPs during the cold seasons account for only 4% of those in the other seasons. There were no significant decreases in warm-temperature INP concentrations after heat treatment for samples collected from January to early April 2022 and December 2022 to March 2023, indicating that biological INPs concentrations were very low during the cold seasons. In the other seasons (April–November), many samples showed spectra in which heat treatment significantly decreased INP concentrations at temperatures > −15 °C and almost no change at lower temperatures, suggesting abundant heat-labile INPs, presumably of biological origin.

## 3.6 Relationships among INPs types inferred from treatments

After $H_2O_2$ treatment, almost all samples showed significant decreases (83–97%, with an average reduction of 91%)) in INP concentrations across all measured activation temperatures (Figure 7 and Figure S2), suggesting substantial contributions from organic INPs through the entire temperature spectrum. The inorganic INPs and other organic INPs were correlated at −15 °C and −25 °C (Figure 8e and 8f), especially at the lower temperature. The correlations clustered around the 10:1 line, with ratios of 14 and 13 at −15 °C and −25 °C, respectively. These results indicated a dominant contribution from organic components suggestive of a soil origin, as previous studies found that soils contain abundant organic INPs in addition to mineral INPs (Tobo et al., 2014; O'Sullivan et al., 2014; Pereira et al., 2022; Suski et al., 2018; Testa et al., 2021). A short-term observational study (DeMott et al., 2025) at the Storm Peak Laboratory in the Colorado Rockies also attributed soil INPs as the dominant INP source in late summer and early fall. These

correlations provide further support that the coarse dust factor was mainly from resuspension of local or regional soil dust. Testa et al. (2021) also found a correlation at −25 °C between other organic and inorganic INPs, with a mean ratio of 5.5 in samples from north-central Argentina. Our results and those of Testa et al. (2021) suggest that other organic INPs and inorganic INPs are co-emitted from similar sources. The correlation coefficient increased at lower temperatures, possibly because other organic and inorganic INPs had limited contributions at −15 °C, a range strongly influenced by biological aerosols. Note that INPs from biomass burning could also contribute to this correlation, as they show similar organic and inorganic INP characteristics in their INP spectra (Schill et al., 2020; Barry et al., 2021b). The only exception is sample 2021-10-14, for which similar INP concentrations were observed in the base, heat, and $H_2O_2$ treatments at temperatures $< -15$ °C, suggesting that it was likely dominated by mineral dust.

Unexpectedly, a correlation was observed between the biological/heat-labile INPs and other organic INPs at −15 °C ($R^2 = 0.61$, n = 29, Figure 8a). This correlation suggests that increases in heat-labile INPs, presumably biological INPs, were accompanied by increases in organic INPs. This is possibly because both biological and organic INPs during SAIL originated from the same source at this temperature, which was likely soil dust. However, at a warmer temperature (−12.5 °C), the correlation was weaker ($R^2 = 0.10$, n = 20), possibly because biological INPs from non-soil sources contributed at higher temperatures (Huang et al., 2021), as the ratios between biological/heat-labile INPs and other organic INPs changed from 13 (−15 °C) to 43 (−12.5 °C). At −25 °C, correlations between biological/heat-labile INPs and other organic INPs were much weaker, and the ratio was much lower (mean ratio of 2).

### 3.7 Prediction of INPs in the Rocky Mountain region

In this study, $n_s$ values were found to be close to those used in the parameterizations derived from
studies of agricultural soil (Tobo et al., 2014), desert dust (Niemand et al., 2012), and fertile soil
(O'Sullivan et al., 2014) (Figure S14a). Applying the parameterization from Tobo et al. (2014)
using the surface area of particles > 500 nm reasonably estimated INP concentrations (Figure
S14b), but tended to underestimate them. This is possibly because the equation provided by Tobo
et al. (2014) requires an estimation of the surface area of soil particles, which introduces
uncertainties when using $S_{m,500}$; accurately estimating surface area of only soil particles in
atmospheric samples requires additional analysis. Also, the equation from Tobo et al. (2014) is not
valid for T > −18 ℃. The parameterization from O'Sullivan et al. (2014) includes a wider
temperature range, while it tended to underestimate in temperatures warmer than −20 ℃ and
overestimate at colder temperatures (Figure S14b). The parameterization from Niemand et al.
(2012) was derived from desert dust, consisting mainly of mineral dust. Some of their dust samples
might have contained arable dust or biological INPs (Beall et al., 2022). Application of their
parameterization to our data showed a bias at warm temperatures (Figure S14b). Therefore, we
developed a parameterization based on the SAIL samples, which could be useful for estimating
atmospheric INPs dominated by organic-containing soil dust.
Parameterization was based on an assumed relationship between IN active surface site density ($n_s$,
$m^{-2}$) calculated from surface area concentrations of particles > 500 nm ($S_{m,500}$) and temperature
(T, ℃) over all seasons. First, a single polynomial equation fitted to data from all samples was
obtained (Figure S15), and the INP concentrations predicted from this equation and the measured
$S_{m,500}$ were compared with the measured INP concentrations (Figure S15). The predicted INPs
were mostly within one order of magnitude of the measured values, suggesting this fit generally
provides a reasonable estimation of INPs. However, the predicted INPs showed an overestimation
trend for some samples around −15 °C. This is due to the limited contribution of biological/heat-
labile INPs in cold seasons.
To better represent the INP temperature spectra accounting for the seasonality of biological
contributions, parameterization equations were fitted as polynomials for samples from other
seasons (April–November, equation 1) and cold seasons (December–March, equation 2) separately
(Figure 9), as follows,

$$\ln (n_s) = -0.007T^2 - 0.785T + 6.636 \ (-30 \ ^\circ\text{C} < \text{T} < -6 \ ^\circ\text{C}) \qquad (1)$$

$$\ln (n_s) = -0.030T^2 - 1.833T - 5.076 \ (-20 \ ^\circ\text{C} < \text{T} < -10 \ ^\circ\text{C}) \qquad (2)$$

Equation 1 was derived for most samples except those from cold seasons, and Equation 2 was
obtained for samples from cold seasons (December–March). Equation 2 was constrained to
intersect with Equation 1 at −20 °C, as there were no significant differences at T < −20 °C, and
Equation 1 was used in this range. The agreement between predicted and measured INPs based on
these two equations showed improvement compared to using the one equation method above
(Figure S15), better representing the measured INPs across the full measured temperature range
(Figure 9). The accurate prediction of the nearly two years of observations of INPs using this single
parameterization, rather than requiring multiple $n_s$-based parameterizations for specific sources, is
likely due to the dominance of coarse dust as the primary INP source in this region. Note that the
lack of precise parameterization for biological INPs still introduces larger uncertainties in
predictions at warm temperature ranges, while applying the two-equation parameterization based
on the seasonal signature of biological INPs identified in this study improves the prediction.

**4 Summary and atmospheric implications**
This study comprehensively characterized INPs over a nearly two-year period in the mountainous
Upper Colorado River Basin region. The observed average INP concentrations were 0.15 $L^{-1}$ at
$-15$ °C and 16 $L^{-1}$ at $-25$ °C. Clear seasonal variations of INP concentrations and temperature
spectra were observed, with low concentrations in winter, increasing in spring, peaking in summer,
decreasing in autumn, and returning to low levels in winter. The aerosol types in this region were
identified as coarse dust, fine dust, biomass burning, sulfate-dominated, and nitrate-dominated.
Coarse dust concentrations were strongly correlated with INP concentrations in all seasons and
over a large temperature range, suggesting that background INPs in the study region were strongly
influenced by coarse dust. Further analysis of the IN active surface density site supported the
dominant role of coarse dust in INPs. Abundant organic INPs were identified, suggesting that
organic-containing soil dust was the primary source of INPs. Back trajectories showed that coarse
dust mostly originated from local or regional sources. This study also found clear evidence of
biological and heat-labile INPs, which showed strong seasonal dependence. Heat-labile,
presumably biological INPs were present during warm seasons but were significantly decreased
in winter. Two parameterization equations based on IN active surface site density were developed
for warm and cold seasons separately. These equations well estimated the measured INP
concentrations across the measured temperature and concentration ranges. The parameterization
developed here could be useful for representing INPs from organic-containing soil dust in other
mountain regions.
This long-term observation identified that organic-containing soil dust is the major source of INPs
in the Rocky Mountain region. Biomass burning aerosols and fine dust, likely from long-range
transport, play less important roles in INPs compared to coarse dust in this region. Our results
indicate that soil dust from nearby regions (e.g., the Colorado Plateau and Central Rockies), rather
than long-range transported fine and mineral dust, dominates contributions to the INPs in the
Upper Colorado River Basin, and therefore influences aerosol-cloud interaction and precipitation.
This study investigated the major INP sources by linking long-term INP measurements with
aerosol source apportionment, presenting reasonable results that agree well with other studies
(DeMott et al., 2025; Lacher et al., 2025). This approach may have broader applicability for INP
source attribution in other regions. Future studies combining short-term online INP measurement
(e.g., continuous flow diffusion chamber, CFDC) with detailed chemical analyses (e.g., mass
spectrometer) would be valuable for providing more direct evidence of INP sources, such as
demonstrated by the approaches used in Cornwell et al. (2023, 2024).
In this study, the origins of biological INPs are unclear due to the limitations of the analytical
methods used. One possibility is that biological INPs were associated with soil dust, e.g., fungi in
soil (Conen and Yakutin, 2018; O'Sullivan et al., 2016). This is supported by the correlation
between biological/heat-labile INPs and other organic INPs at −15 °C. Also, raindrop impact could
be an important biological INP emission pathway (Prenni et al., 2013; Mignani et al., 2025). In
cold seasons, snow covers the Rocky Mountain region and inhibits the suspension of local soil
dust, which is considered as the main source of INPs in this region. This assumption that coarse
dust emissions are suppressed in cold seasons is also supported by the observation that more than
half of the samples in cold seasons were dominated by mixed sources and fine dust, instead of
coarse dust. However, the correlation between biological/heat-labile INPs and other organic INPs
was weaker ($R^2 = 0.10$) at a warmer temperature (−12.5 °C). This suggests that there are
contributions from other biological INPs (e.g., those originating from vegetation) at warmer
freezing temperatures, which are likely independent of soil dust emissions, thereby complicating
the estimation of biological INPs. For example, airborne bacteria (Bowers et al., 2012) and pollen
(Fall et al., 1992) in the Rocky Mountain region decrease in winter and increase during the warmer
seasons. This seasonal pattern is consistent with the variation in biological/heat-labile INPs
observed in this study, suggesting that these biological particles may represent potential sources of
the biological INPs. In this case, the INP spectra in cold seasons likely represent that of soil dust,
and the spectra in other seasons represent the combination of soil dust and biological INPs. While
bacteria may also be partly related to soil, as summer bacteria taxa were reported to likely originate
from soil and leaf surface (Bowers et al., 2012), the specific ice-active taxa among them require
further investigation. In future studies, identifying the most abundant biological INPs in this region
and determining whether they originate from vegetation, soil-associated sources, or from a
combination of both would help improve our understanding of biological INP variability and
improve their estimation. However, besides heat treatment, current approaches provide only
indirect evidence of biological INPs (Cornwell et al., 2023 and 2024; Sanchez-Marroquin et al.,
2021). Comprehensive characterization of biological aerosol types and abundance, or developing
new analytical approaches, would be highly beneficial for advancing biological INP research.
The INP concentrations in this study were similar to those observed in ambient samples influenced
by agricultural soil in Argentina (Testa et al., 2021). The $n_s$ values reported for agricultural soil
samples (Tobo et al., 2014) were higher than those calculated from total surface area (Figure S10),
but similar to $n_s$ when based on surface area of particles > 500 nm (Figure S14). This finding
suggests that using particles > 500 nm is a reasonable threshold for excluding most aerosol types
that are inefficient INP sources, and approximating aerosol surface area contributions from
organic-containing soils. Since SAIL was conducted at a remote site in the Rocky Mountains, these
comparisons suggest that the parameterization developed in this study can potentially be applied
to other remote continental areas. A recent global modeling study (Herbert et al., 2025) found that
including organic INP components in dust particles, which are present in many soils, can
significantly improve predictive accuracy. It is therefore essential to validate the INPs originating
from organic-containing soil dust, beyond the mineral dust that has already been intensively
studied, through field measurements across different continental regions.

**Data Availability**
The data from the SAIL campaign used in this study are available through the ARM Data
Discovery (https://adc.arm.gov/discovery/). This includes INPs data (Shi et al., 2025), source
apportionment results (Zhou et al., 2025a), back trajectory data (Zhou et al., 2025b), merged
aerosol number-size distribution (Zhou et al., 2025c), and particle number size distribution data
from SMPS (Kuang et al., 2024) and OPC (Cromwell et al., 2024).

**Author contributions**
RP, SK, PD, and RZ conceptualized the study. DJ, RZ, KA, and OD performed additional
immersion-mode INP measurements. RZ, RP and SK conducted the source apportionment and
merged the size distribution data. RZ, RP, KA and SK performed the back trajectory analysis. KB
performed qPCR analysis. RZ wrote the manuscript, with revisions from RP, PD, and SK. All
authors reviewed the manuscript and contributed to the final version of the manuscript.

**Competing interests**
The authors declare no competing interests.

## Acknowledgements

This work is supported by DOE Atmospheric Systems Research award DE-SC0024202 and DE-
SC0021116. We thank the DOE ARM INP Mentors Jessie Creamean, Tom Hill, and Carson Hume
for coordinating the sample collections and for helping with the combined INP data archival. We
also thank Ty Johnson for assistance with generating the NOAA Hazard Mapping System plots.
The authors acknowledge the NOAA Air Resources Laboratory (ARL) for the provision of the
HYSPLIT transport and dispersion model used in this publication. We also thank the Interagency
Monitoring of Protected Visual Environments (IMPROVE) network for providing aerosol
composition data. IMPROVE is a collaborative association of state, tribal, and federal agencies,
and international partners. US Environmental Protection Agency is the primary funding source,
with contracting and research support from the National Park Service. The Air Quality Group at
the University of California, Davis is the central analytical laboratory, with ion analysis provided
by Research Triangle Institute, and carbon analysis provided by Desert Research Institute. We
thank North American Weather Consultants Inc. for providing records of their cloud seeding
activities near the SAIL campaign site. Kelton Ayars and Oren Dutton would like to acknowledge
support from the Scott Undergraduate Research Experiences program at Colorado State University.

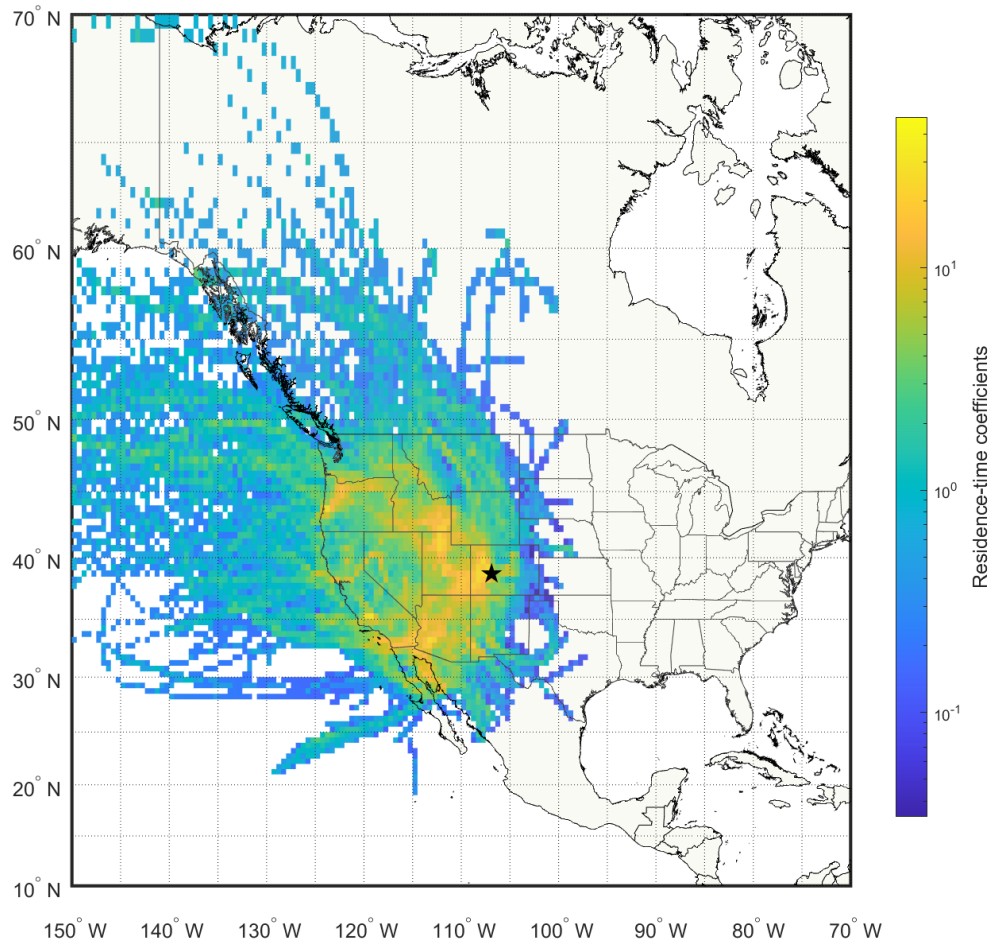


**Figure 1.** Residence-time weighted back trajectories for all sampling periods. 96-hour back trajectories were generated hourly during each sampling period and normalized by residence time and distance from the sampling site. The residence-time coefficients indicate the relative time air masses spent within each grid cell. The black star marks the SAIL sampling location.

733

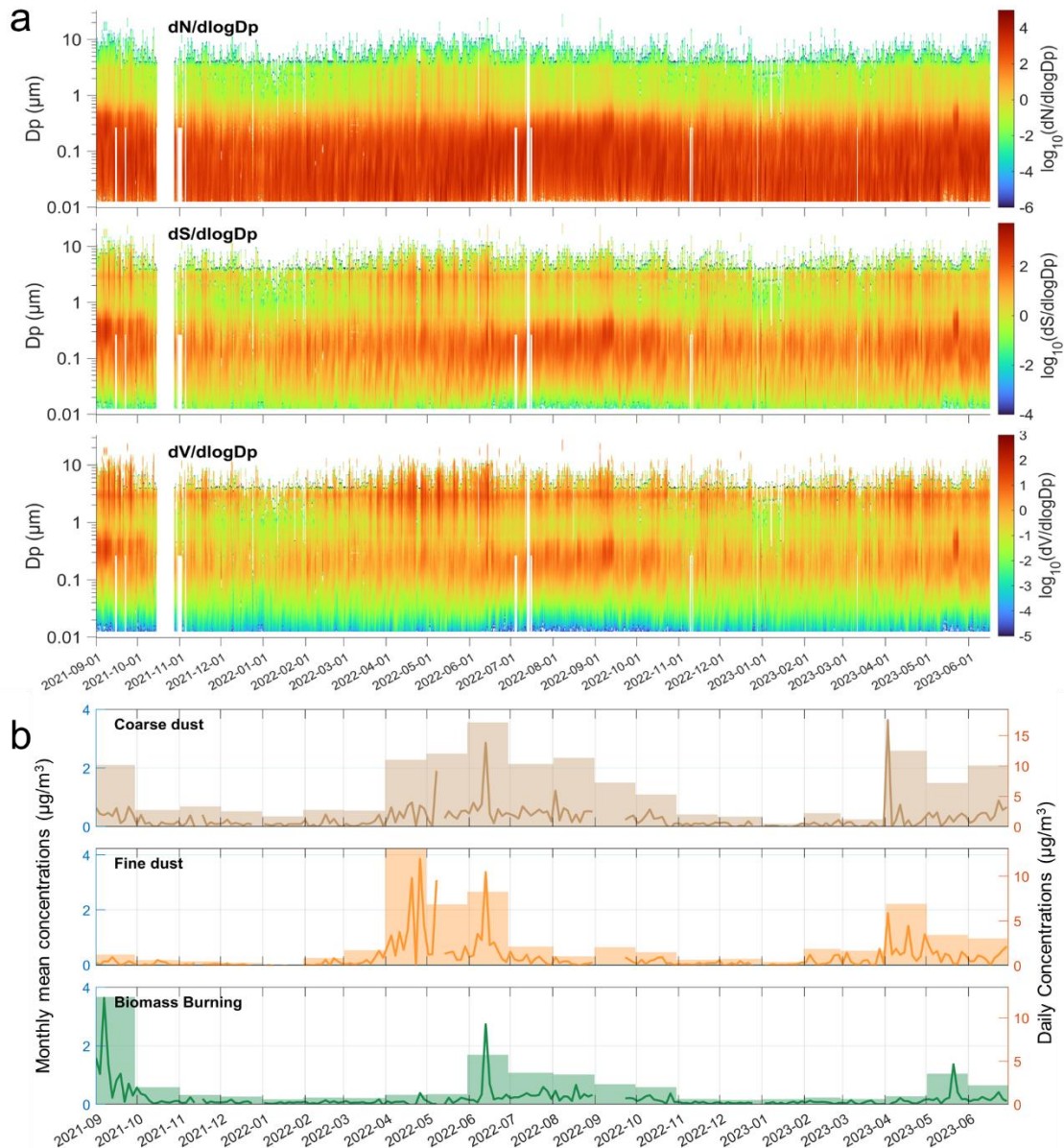

734

**Figure 2.** (a) Number size distribution ($dN/d\log D_p$, scm$^{-3}$), surface area size distribution ($dS/d\log D_p$, μm$^2$/scm$^{-3}$), and volume size distribution ($dV/d\log D_p$, μm$^3$/scm$^{-3}$), derived from merged size distribution assuming spherical particles during the SAIL campaign. (b) Time series of aerosol mass concentrations (μg m$^{-3}$) of coarse dust, fine dust, and biomass burning. Lines and bars represent daily and monthly mean concentrations, respectively.

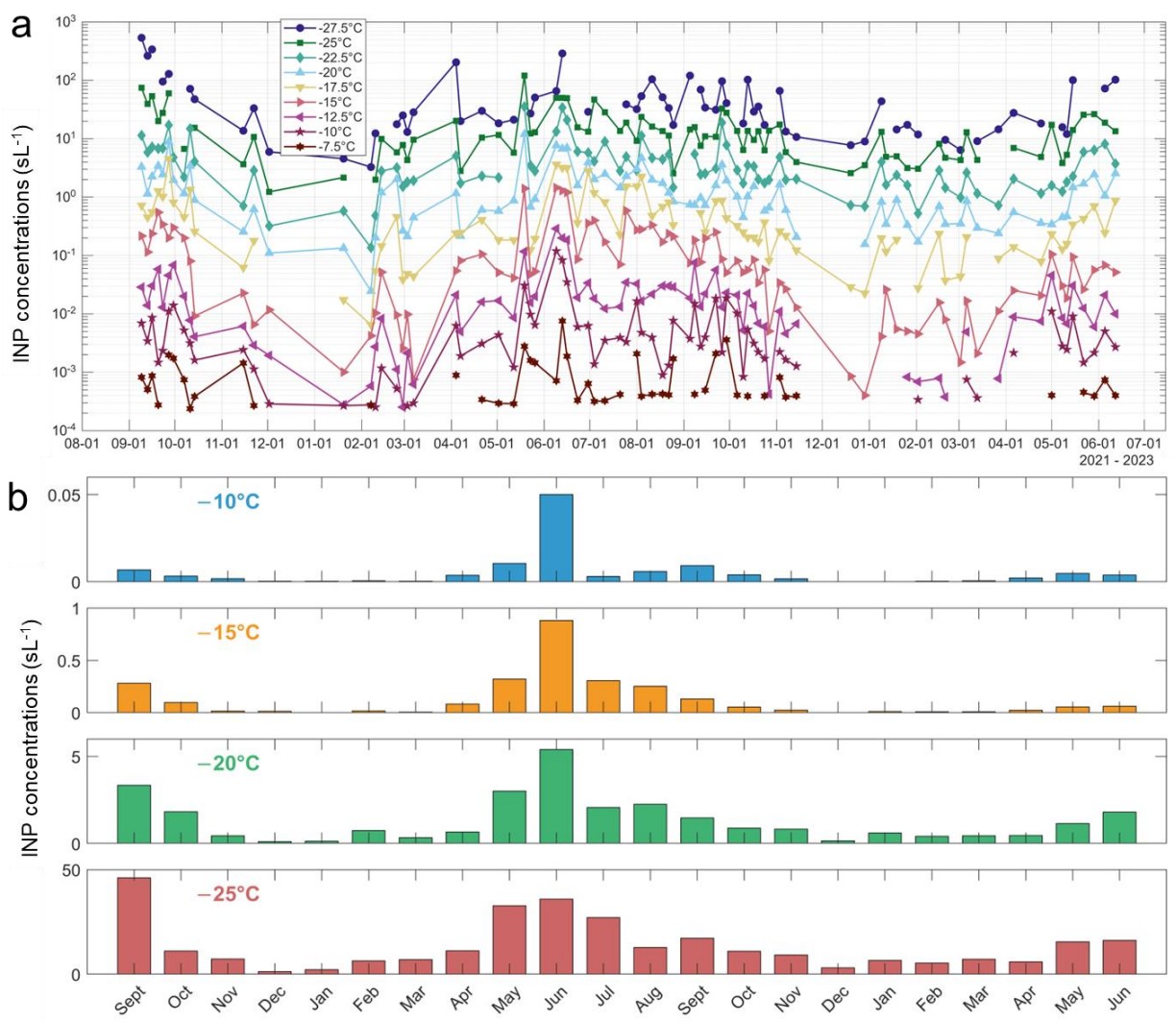

740

**Figure 3.** Measured INP concentrations (sL⁻¹) during the SAIL campaign (September 2021–June
2023), excluding samples obtained during cloud seeding and snowmaking activities. (a) INP
concentrations at temperatures from −7.5 °C to −27.5 °C in 2.5°C intervals. (b) Monthly mean
INP concentrations at temperatures of −10 °C, −15 °C, −20 °C, and −25 °C.


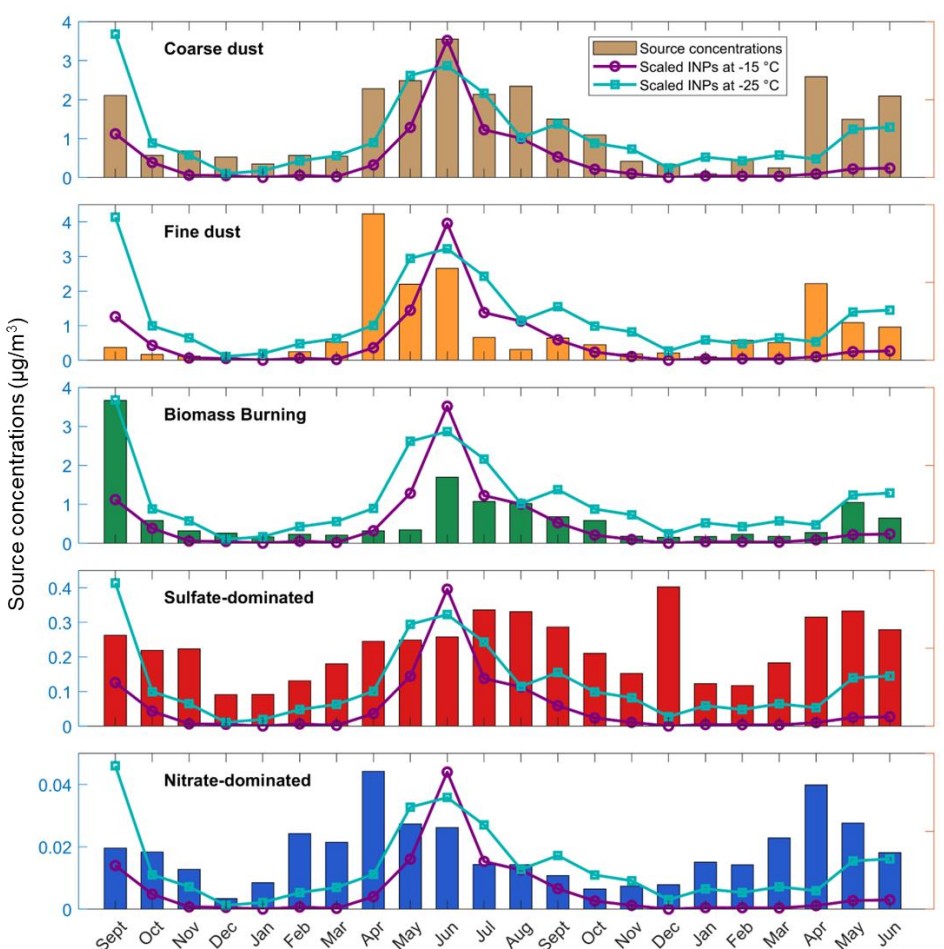


**Figure 4.** Monthly mean INP concentrations (lines, arbitrary scaling) at −15 °C and −25 °C and monthly mean aerosol mass concentrations (bars) from coarse dust, fine dust, biomass burning, sulfate-dominated, and nitrate-dominated sources.





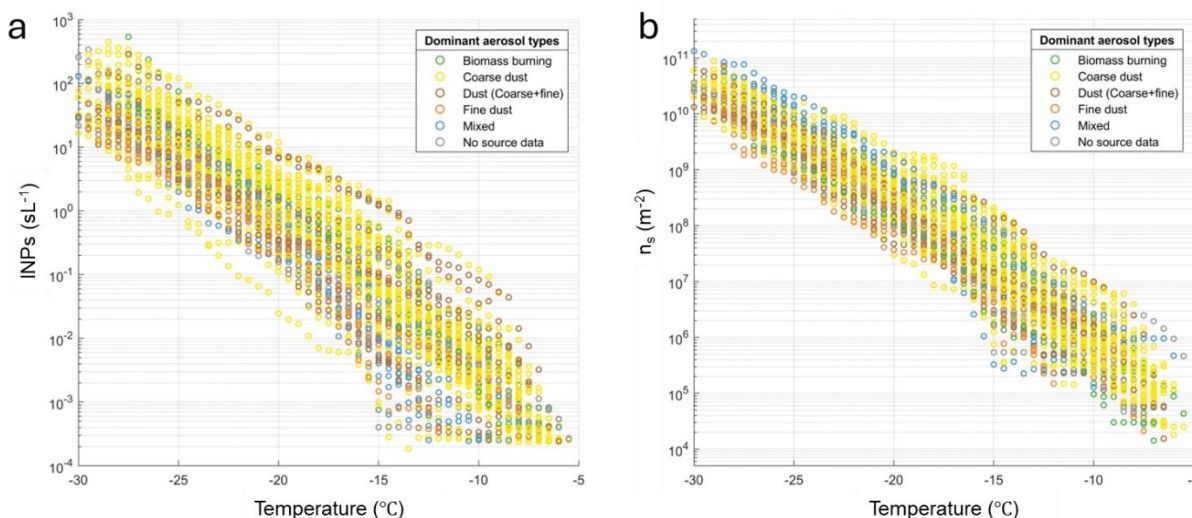


**Figure 5.** (a) Total INP concentration (sL$^{-1}$) temperature spectra, and (b) IN active surface site density ($n_s$, m$^{-2}$) calculated based on the surface area of particles larger than 500 nm. All samples were categorized by the dominant aerosol sources. Colors in the legend represent the dominant aerosol types during sampling.


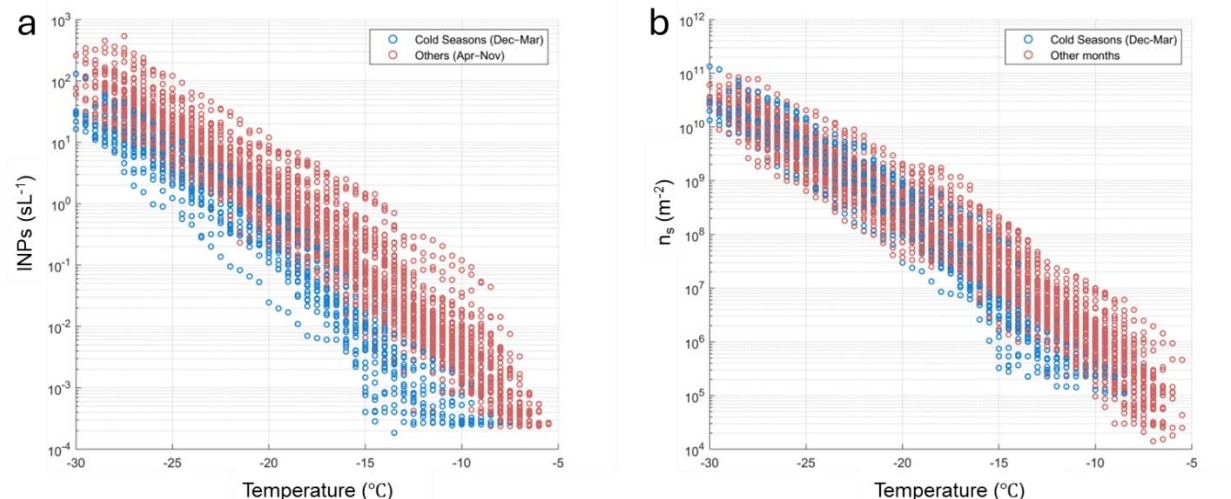


**Figure 6.** (a) INP temperature spectra and (b) IN active surface site density ($n_s$) categorized by sampling date as cold seasons (December–March) and other seasons (April–November). $n_s$ was calculated based on the surface area of particles larger than 500 nm.


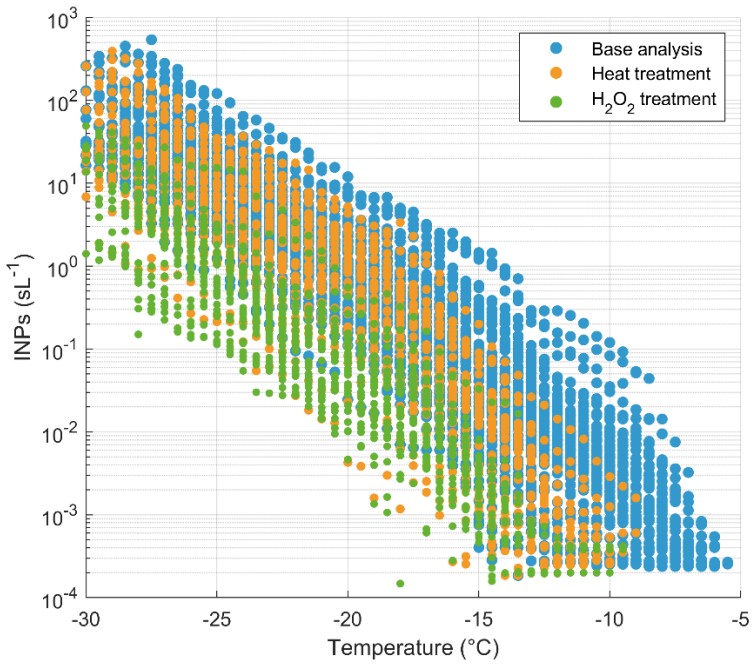


**Figure 7**. Comparison of INP temperature spectra from base (untreated), heat-treated, and $H_2O_2$-
treated analyses.

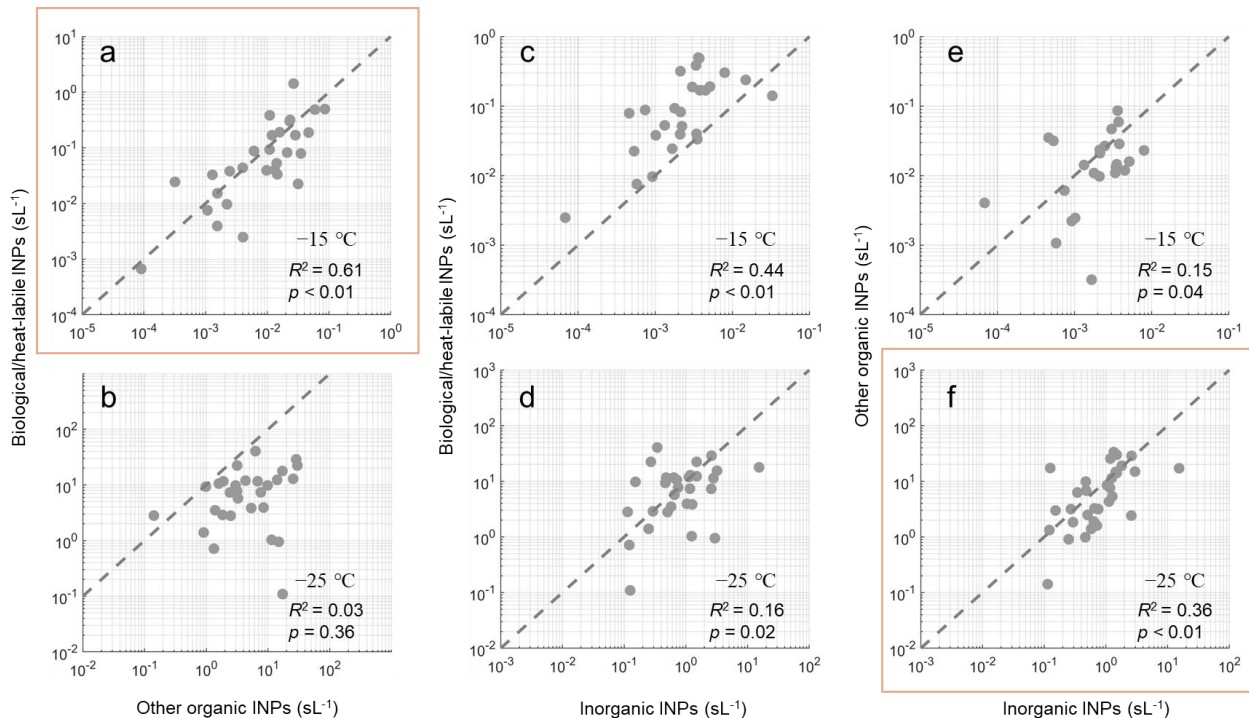

**Figure 8.** Correlations between concentrations of (a, b) biological/heat-labile INPs and other organic INPs, (c, d) biological/heat-labile INPs and inorganic INPs, and (e, f) other organic INPs and inorganic INPs, active at either −15 °C (upper row) or −25 °C (lower row). Dashed lines indicate a 10:1 relationship for reference, and the orange rectangles highlight the strongest correlations at each temperature.

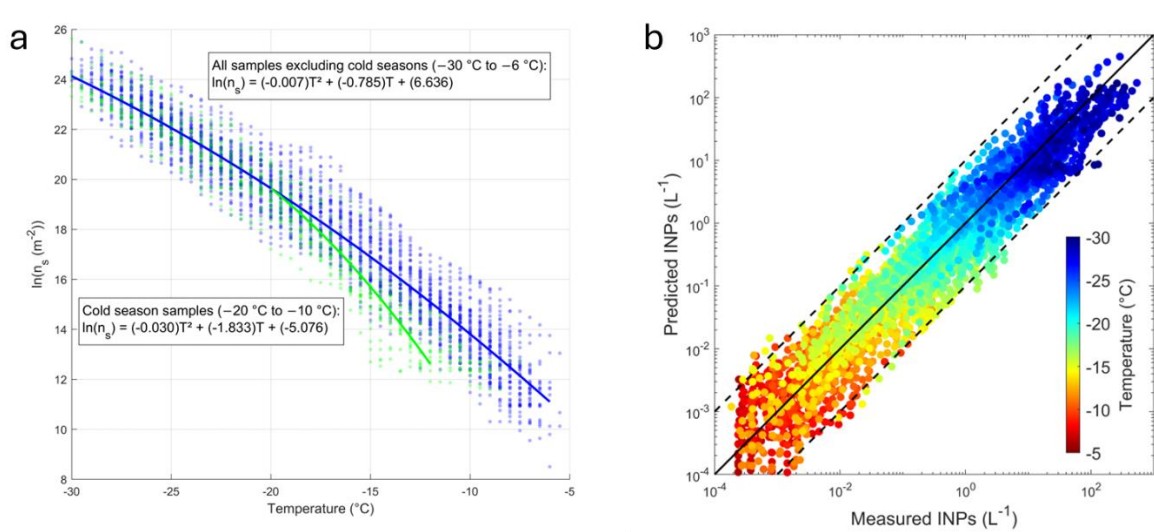

772

**Figure 9.** (a) All cold season (December–March) and other seasons (April–November) INP data, expressed as IN active surface site density $n_s$ (based on the surface area of particles larger than 500 nm), with parameterization fits. All samples, excluding those from cold seasons, were used to develop the parameterization equation for temperatures from −30 °C to −6 °C. A separate equation was developed for samples from cold seasons at temperatures from −20 °C to −10 °C. (b) Comparison between predicted INP concentrations based on the parameterization equations and measured surface area concentrations, and measured INP concentrations.


**Table 1.** Pearson correlation coefficients ($R^2$) between monthly means of INP concentrations and
source factor concentrations.

| | Coarse dust | Fine dust | Biomass burning | Sulfate-dominated | Nitrate-dominated | Coarse dust and biomass burning |
|---|---|---|---|---|---|---|
| -10 ℃ | 0.430* | 0.192 | 0.157 | 0.070 | 0.036 | 0.389* |
| -15 ℃ | 0.568* | 0.175 | 0.316* | 0.093 | 0.040 | 0.590* |
| -20 ℃ | 0.579* | 0.117 | 0.484* | 0.121 | 0.040 | 0.706* |
| -25 ℃ | 0.500* | 0.100 | 0.688* | 0.147 | 0.051 | 0.763* |

*$p < 0.01$

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
