# Peer review of "Seasonal variability, sources, and parameterization of ice-nucleating particles in the"

_EGUsphere, 2025_

## Author Comment (AC1)

This study provides a thorough characterization of airborne INPs from a remote, alpine area at the Mt. Crested Butte study site in the Rocky Mountains. The long-term monitoring of INPs (almost 2 years) allowed for the emergence of trends and conclusions that can only be made from such a comprehensive data set. The study revealed a distinct seasonal variation in INP concentrations with a peak in the summer time. Further, different INP concentrations were correlated with various sources, with organic-containing soil dust dominating the INP population in this area. The parameterizations developed can be useful for predicting INPs in remote continental regions. This paper adds a valuable dataset to the field of ice nucleation and will be of interest to ACP readers. It is recommended that this paper should be accepted for publication after the authors address some minor revisions.

We appreciate the reviewer's positive assessment of our study and the valuable comments. We have carefully revised the manuscript in response to all suggestions.

**General comments:**

The discussion on the influence of vegetation on the INP population could benefit from additional context and references. Could the authors expand on this a bit more? For example, did the authors consider the influence of pollen? Biological INPs were present in warm seasons and decreased in winter (line 612). Does that line up with the pollen season? Furthermore, it might be helpful if the authors elaborated a bit more on the possible source pathways that link vegetation and soil, since the main conclusion is that organic-containing soil was the dominant INP.

For pollen in the Rocky Mountain region, Fall et al. (1992) found that pollen during the cold season (October to May) accounted for only about 18% of the annual influx, and the remaining 82% occurred during the warmer months. This pattern is consistent with the seasonal variation pattern in biological/heat-labile INP observed in this study, suggesting the possibility that pollen contributes to the biological INPs in the Rocky Mountains.

Bowers et al. (2012) investigated airborne bacteria at the Storm Peak Laboratory, located in the Rocky Mountains, and found that bacterial abundance was lower in winter and increased in fall and spring, indicating that bacteria may also serve as a source of biological INPs in this region. They also found that summer bacteria taxa were likely derived from soil and leaf-surface environments, suggesting that bacteria-related INPs may partly originate from soil dust.

Also, rainfall was suggested as an important source pathway of biological INPs (Prenni et al., 2013; Mignani et al., 2025).

The seasonal pattern provides indirect evidence for the possible sources of biological INPs, and further studies are required to identify the ice-active types of biological aerosols.

To further discuss these points, the following sentences have been added:

"Also, raindrop impact could be an important biological INP emission pathway (Prenni et al., 2013; Mignani et al., 2025)." (Lines 655–656)

"For example, airborne bacteria (Bowers et al., 2012) and pollen (Fall et al., 1992) in the Rocky Mountain region decrease in winter and increase during the warmer seasons. This seasonal pattern is consistent with the variation in biological/heat-labile INPs observed in this study, suggesting that these biological particles may represent potential sources of the biological INPs." (Lines 665–669)

"While bacteria may also be partly related to soil, as summer bacteria taxa were reported to likely originate from soil and leaf surface (Bowers et al., 2012), the specific ice-active taxa among them require further investigation." (Lines 670–673)

Four related references have been added.

"Bowers, R. M., McCubbin, I. B., Hallar, A. G., and Fierer, N.: Seasonal variability in airborne bacterial communities at a high-elevation site, Atmos. Environ., 50, 41-49, https://doi.org/10.1016/j.atmosenv.2012.01.005, 2012."

"Fall, P. L.: Spatial patterns of atmospheric pollen dispersal in the Colorado Rocky Mountains, USA, Review of Palaeobotany and Palynology, 74, 293-313, https://doi.org/10.1016/0034-6667(92)90013-7, 1992."

"Mignani, C., Hill, T. C. J., Nieto-Caballero, M., Barry, K. R., Bryan, N. C., Marinescu, P. J., Dolan, B., Sullivan, A. P., Hernandez, M., Bosco-Lauth, A., van den Heever, S. C., Stone, E. A., Grant, L. D., Perkins, R. J., DeMott, P. J., and Kreidenweis, S. M.: Ice-Nucleating Particles Are Emitted by Raindrop Impact, J. Geophys. Res: Atmos., 130, https://doi.org/10.1029/2024jd042584, 2025."

"Prenni, A. J., Tobo, Y., Garcia, E., DeMott, P. J., Huffman, J. A., McCluskey, C. S., Kreidenweis, S. M., Prenni, J. E., Pöhlker, C., and Pöschl, U.: The impact of rain on ice nuclei populations at a forested site in Colorado, Geophys. Res. Lett., 40, 227-231, https://doi.org/10.1029/2012gl053953, 2013."

**PMF and source apportionment: While some atmospheric scientists are very familiar with PMF, others may not be fully convinced by your claims without having prior knowledge of PMF. Therefore, including a short explanation of PMF targeted for non-experts would make this section more convincing.**

An explanation of the PMF model has been added as follows:

"PMF is a receptor model that decomposes an observation matrix into factor profiles and their corresponding contributions. These factors are related to emission sources and/or atmospheric

processes, providing a quantitative assessment of source influences (Paatero and Tapper, 1994)."
(Lines 202–206)

Additional information on the PMF procedure and the identification of PMF factors have been added as follows:

"A five-factor solution was selected as the optimal solution based on the $Q/Q_{exp}$ value and interpretation of the physical meanings of the factors (Brown et al., 2015). The corresponding factor profiles and time series are shown in Figures S5 and S6. These factors were identified, based on chemical signatures and previous literature, as coarse dust, fine dust, biomass burning, sulfate-dominated, and nitrate-dominated sources. Coarse and fine dusts had high contributions from Al, Ca, Fe, Mg, and Si, which are the main components of mineral dust (Liu and Hopke, 2003). Coarse dust explained more than 90% of the coarse mass (> $PM_{2.5}$), while there was no contribution from coarse mass in the fine dust factor. The biomass burning factor was strongly associated with organic and elemental carbon, which are mainly from combustion processes, and K, a tracer of biomass burning (Hopke et al., 2020). The other two factors are dominated by nitrate and sulfate, which are related to the formation of secondary aerosols and possibly some primary emissions from regional sources that include energy production and distant urban regions. Some similar factors were also resolved in published PMF analyses using IMPROVE data (Liu and Hopke, 2003; Hwang and Hopke, 2007)." (Lines 219–232)

**This may stem from my lack of PMF knowledge, but it was not clear to me how sample classification rules were determined. For example, why was 40% chosen as the cutoff for coarse dust contribution to $PM_{10}$?**

Thank you for this comment, which led us to better explain how we used PMF to classify air mass types. The PMF results showed the fractional contributions of each aerosol type to every sample, and these proportions were used to identify the dominant aerosol sources for each sample. This classification step is independent of the PMF analysis itself. A figure (Figure S8) showing the proportions of source contribution for all samples has been added to the SI to better illustrate how dominant sources were assigned. As shown in Figure S8, each sample was influenced by a variety of sources, and thus identifying a cutoff to assign the major source was somewhat arbitrary. Generally, a source contributing more than 50% is regarded as the dominant source. For coarse dust, if it contributed more than 50% to the total $PM_{10}$ mass concentration or contributed more than 40% and was higher than other sources, this sample was categorized as a coarse dust-dominated sample.

To clarify the categorization, the method has been updated as follows:

"(1) Coarse dust, if coarse dust contributed more than 50% to the total $PM_{10}$ mass concentration or contributed more than 40% and represented the largest contribution among all sources."
(Lines 240–242)

The figure for proportions of source contributions:

[Figure]

**Figure S8.** Proportions of source contributions for all samples. The label below each sample indicates its classification: CD (coarse dust), B (biomass burning), D (dust), FD (Fine dust), M (Mixed sources), and N (no source information).

**Ulbrich et al., 2009 provides useful PMF guidelines when working with AMS data. Can you provide a similar reference for the technique that was used for the IMPROVE dataset? This will be very useful for scientists who want to learn PMF and reproduce your work, especially considering the interpretation of factors and if that is approached differently for different techniques.**

We have added the following references that have applied the PMF model to IMPROVE datasets for source apportionment. We also note that the version of PMF we use was developed by the EPA in the 1990s for source apportionment of environmental datasets and continues to undergo further refinement: https://www.epa.gov/air-research/positive-matrix-factorization-model-environmental-data-analyses:

Liu and Hopke (2003) is an early work that applied PMF to resolving the source of $PM_{2.5}$ at two high elevation IMPROVE sites in the western U.S.

Hwang and Hopke (2007) investigated the sources at a west coast IMPROVE site using the PMF model.

Brown et al. (2015) provide methods for evaluating the results using EPA PMF 5.0 with examples.

These studies describe the PMF model and its application to IMPROVE datasets, and they resolved some similar factors as we found in this study, which support our source apportionment results.

These points have been added in manuscript as follows:

"A five-factor solution was selected as the optimal solution based on the $Q/Q_{exp}$ value and interpretation of the physical meanings of the factors (Brown et al., 2015). The corresponding factor profiles and time series are shown in Figures S5 and S6. These factors were identified, based on chemical signatures and previous literature, as coarse dust, fine dust, biomass burning, sulfate-dominated, and nitrate-dominated sources. Coarse and fine dusts had high contributions from Al, Ca, Fe, Mg, and Si, which are the main components of mineral dust (Liu and Hopke, 2003). Coarse dust explained more than 90% of the coarse mass ($> PM_{2.5}$), while there was no contribution from coarse mass in the fine dust factor. The biomass burning factor was strongly associated with organic and elemental carbon, which are mainly from combustion processes, and K, a tracer of biomass burning (Hopke et al., 2020). The other two factors are dominated by nitrate and sulfate, which are related to the formation of secondary aerosols and possibly some primary emissions from regional sources that include energy production and distant urban regions. Some similar factors were also resolved in published PMF analyses using IMPROVE data (Liu and Hopke, 2003; Hwang and Hopke, 2007)." (Lines 219–232)

"Brown, S. G., Eberly, S., Paatero, P., and Norris, G. A.: Methods for estimating uncertainty in PMF solutions: examples with ambient air and water quality data and guidance on reporting PMF results, Sci. Total Environ., 518-519, 626-635, https://doi.org/10.1016/j.scitotenv.2015.01.022, 2015."

"Hwang, I. and Hopke, P. K.: Estimation of source apportionment and potential source locations of PM2.5 at a west coastal IMPROVE site, Atmos. Environ., 41, 506-518, https://doi.org/10.1016/j.atmosenv.2006.08.043, 2007."

"Hopke, P. K., Dai, Q., Li, L., and Feng, Y.: Global review of recent source apportionments for airborne particulate matter, Sci. Total Environ., 740, 140091, https://doi.org/10.1016/j.scitotenv.2020.140091, 2020."

"Liu, W. and Hopke, P. K.: Origins of fine aerosol mass in the western United States using positive matrix factorization, J. Geophys. Res: Atmos., 108, https://doi.org/10.1029/2003jd003678, 2003."

**In addition, could the authors please mention the techniques used for the filter and elemental analysis that were used from the IMPROVE network.**

The techniques used by the IMPROVE network for chemical analysis are introduced in the IMPROVE Data User Guide 2023 (Version 2), and have been added as follows:

"In the IMPROVE program, elemental analysis was performed on the Teflon filters using X-ray fluorescence (XRF), water-soluble anions were analyzed by ion chromatography (IC), and elemental carbon (EC) and organic carbon (OC) were analyzed using a thermal-optical carbon analyzer (Hand, 2023)." (Lines 210–213)

"Hand, J.: IMPROVE Data User Guide 2023 (Version 2), Interagency Monitoring of Protected Visual Environments (IMPROVE) Program, available at: https://vista.cira.colostate.edu/Improve/data-user-guide/, 2023."

**Just recently, a preprint by Lacher et al. 2025 was released, which also studied INP concentrations in the U.S. Rocky Mountains. The sampling period from Lacher et al., 2025 has some overlap with the sampling period in this paper and the sampling sites are close together. It would be very beneficial if you can compare trends in your data set with those in the other study.**

Lacher et al. (2025) found that INP concentrations were lowest in winter and increased in spring, and that supermicron particles were the major contributor to INPs in the Rocky Mountains, based on online INP measurements at cold activation temperatures (−22 °C to −32 °C). These results agree well with our findings. Their results are compared and discussed in the manuscript as follows:

"This is comparable to online INP measurements in the Rocky Mountain region (median: 8.2 L−1 at −26 °C; Lacher et al., 2025)." (Lines 314–315)

"Recent online INP measurements for activation temperatures from −22 °C to −32 °C conducted from October 2021 to May 2022 and January to May 2025 at the Storm Peak Laboratory in the Rocky Mountains (Lacher et al., 2025) found a similar seasonal pattern, with the lowest INP concentrations in winter and increased in spring, suggesting that the INP sources could be similar and may dominate INPs across a broaden region of the Rocky Mountains." (Lines 347–351)

"Furthermore, Lacher et al. (2025) provided direct evidence that INPs active at cold temperatures were significantly contributed by supermicrometer particles, which they attributed to dust, in the Rocky Mountains. Their observation site was located near to the IMPROVE site at Mount Zirkel, where our PMF analyses identified similar sources and trends to those near the SAIL (Text S1), suggesting that INPs in both studies were impacted by coarse dust." (Lines 388–393)

**Specific comments:**

**Title: The paper might benefit from a more precise title. Adding in your main finding into the title would make it more targeted.**

Thanks for the comment, the main finding regarding the importance of soil dust and biological contributions has been incorporated into the title, as follows:

"Seasonal variability, sources, and parameterization of ice-nucleating particles in the Rocky Mountain region: Importance of soil dust and biological contributions"

**Line 363: course dust was found to correlate with INPs at -10 °C (R$^2$ = 0.43). Is this R$^2$ value significant for INP characterization? Is this a typical value you expect in INP field studies? I feel like this value is a bit low to suggest a direct correlation.**

From Table 1 in the manuscript, the p-value of this correlation is lower than 0.01, suggesting that it is statistically significant. However, the $R^2$ is lower than the correlations observed at colder temperatures (0.50-0.58). To clarify this point, the related sentences have been revised as follows:

"Coarse dust showed good correlations with INPs active at all temperatures (Table 1), with correlation coefficients increasing for colder activation temperatures, suggesting that coarse dust is a major source of INPs, particularly at lower temperatures. This is consistent with previous findings that dust dominates the INPs at temperatures below −20 °C (Beall et al., 2022; Testa et al., 2021; Kanji et al., 2017)." (Lines 384–388)

"Interestingly, coarse dust presented a weaker correlation with INPs at −10 °C ($R^2$ = 0.43), a temperature range usually associated with biological INPs. This may be due to the large number of coarse dust particles, biological INPs carried on dust particles, and/or the inclusion of biological particles in the coarse dust factor, as biological particles are mostly supermicron in size (Després et al., 2012)." (Lines 393–397)

**Line 632-636: Can you give some example methods on how this could be done? Comments like these are very useful when planning future studies and suggesting example methods would be very beneficial to the community.**

In addition to the heat treatment method, previous studies have provided indirect evidence of biological INPs based on fluorescence measurement, combining CFDC and mass spectrometer, and microscopy (Cornwell et al., 2023 and 2024; Sanchez-Marroquin et al., 2021). However, from the authors' knowledge, based on currently available methods, it is still very difficult to obtain direct evidence of biological INP identities. Comprehensively investigating the types and abundance of biological aerosols in the Rocky Mountains would be helpful. For directly investigating the biological INPs, new methods that separate INPs, especially biological origins, from other particles need to be developed. To include this information, the sentence has been revised as follows:

"In future studies, identifying the most abundant biological INPs in this region and determining whether they originate from vegetation, soil-associated sources, or from a combination of both would help improve our understanding of biological INP variability and improve their estimation. However, besides heat treatment, current approaches provide only indirect evidence of biological INPs (Cornwell et al., 2023 and 2024; Sanchez-Marroquin et al., 2021). Comprehensive characterization of biological aerosol types and abundance, or developing new analytical approaches, would be highly beneficial for advancing biological INP research." (Lines 673–679)

**Line 153: please include the equation for INP concentration directly in the methods section in addition to your reference to Vali, 1971**

The equation that calculates the cumulative INP concentrations as a function of temperature has been added as follows:

"Cumulative INP concentrations as a function of temperature ($n_{INPs}$(T), INPs per liter of air) were calculated based on the Vali method (Vali, 1971) using:

$$n_{INPs}(T) = ln(\frac{N_0}{N_0 - N(T)}) \times \frac{V_w}{V_c} \times \frac{1}{V_a}$$

where $N_0$ is the total number of wells containing aliquots, $N(T)$ is the cumulative number of wells frozen at temperature $T$, $V_w$ is the volume of water used for particle resuspension, $V_c$ is the aliquot volume added to each well, and $V_a$ is the total sampled air volume." (Lines 148–153)

**Figure 3a: This graph is a little difficult to read. Could the lines be thicker to make it easier to follow? Additionally, the data presented in Figure 3a and Figure S3a are very similar. Is it necessary to show both versions? Please also remove the "a" from Figure S3 as there is only one panel.**

To improve the readability of Figure 3a, the colors, markers, and size of the figure have been modified as follows:

[Figure]

The "a" in Figure S3 has been removed.

Figure S3 presents all samples measured during the SAIL campaign (in total 113 samples). Samples affected by artificial cloud seeding and snowmaking activities are shown with grey shadows in Figure S3, and these samples were not discussed in this manuscript. Figure 3a shows only the 83 samples that were not affected by cloud seeding and snowmaking activities. The difference has been explained in lines 187–190 and in the caption of Figure 3.

**Figure 7: Can you add a line to the figure caption to explain that the y-axis and x-axis is the same for the 2 plots in each column? It was a bit challenging to understand the figure right away.**

A label has been added to each panel, and the caption of Figure 7 (Figure 8 in the revision) has been revised as follows:

"**Figure 8**. Correlations between concentrations of (a, b) biological/heat-labile INPs and other organic INPs, (c, d) biological/heat-labile INPs and inorganic INPs, and (e, f) other organic INPs and inorganic INPs, active at either −15 °C (upper row) or −25 °C (lower row). Dashed lines indicate a 10:1 relationship for reference, and the orange rectangles highlight the strongest correlations at each temperature." (Lines 765–769)

**Figure S1 and S2: Are the black and orange curves in Figure S1 the same as the curves shown in Figure S2? If so, please consider removing S1 as the data is repeated.**

Yes, the black dots and orange dots in both figure S1 and S2 represent the INP spectra of base analyses and heat treatment, respectively. However, Figure S1 shows all samples subjected to heat treatment (43 samples in total), whereas Figure S2 shows only the subset of samples (34 in

total) that underwent $H_2O_2$ treatment. To clarify this difference, the captions of the two figures have been revised as follows:

"Figure S1. The INP temperature spectra of samples that were subjected to heat treatments (43 samples in total). The base analyses (black dots) are shown along with spectra after heat treatment (orange dots). For clarity, uncertainties are not shown here."

"Figure S2. The INP temperature spectra of samples with base analysis (black dot), heat treatment (orange dot), and $H_2O_2$ treatment (blue dot; 34 samples in total). For clarity, uncertainties are not shown here."

**Figure S12: Please add a marker to show the location of the sampling site. Something like what was done in Figures 1 and S8.**

A marker shows the sampling location has been added to Figure S12 (Figure S13 in the revision) as follows:

[Figure]

Figure S13. NOAA Hazard Mapping System products for the sample days, which are dominated by biomass burning aerosols. The black stars mark the SAIL sampling location.

**Technical corrections:**

**Figure S8: Please add a legend or a line in caption to explain what the black star represents.**

The caption of Figure S8 (Figure S9 in the revision) has been revised as follows:

"**Figure S9.** Residence-time weighted back trajectories for samples that were categorized as dominated by (a) coarse dust, (b) fine dust, (c) dust, and (d) biomass burning. The categorization was based on aerosol source contributions derived from the PMF analysis; details of the categorization method are provided in the Methods section. The black stars indicate the SAIL sampling location."

**Figure S10: Please add x-axis label**

The x-axis label "Temperature (°C)" has been added.

**Please ensure all figures are presented with adequate resolution.**

The resolutions of all figures have been checked and are sufficient for publication.

**There is inconsistent figure border formatting throughout the paper. And in some cases, the figure border cuts through text (S3, S4, S6, S7, S9, S13).**

The borders of these figures have been modified.

**Line 210: typo in April**

The typo has been corrected.

**Text S1: link for the IMPROVE sop #351 says page not found. Please update.**

The IMPROVE SOP in Text S1 has been revised as a reference as follows:

"These uncertainties were therefore calculated based on the method introduced in Niño (2021) and the IMPROVE standard operating procedure (SOP 351; IMPROVE, 2021)."

"IMPROVE (2019): IMPROVE Standard Operating Procedure 351: Data Processing and Validation, available at: https://vista.cira.colostate.edu/improve/wp-content/uploads/2019/06/IMPROVE-SOP-351_Data-Processing-and-Validation_06.2019.pdf"

Other revisions

1. Citations including "Mccluskey" corrected as "McCluskey".

2. Sentence in line 242 has been revised as follows:

   "Further details on the PMF analysis and results, as well as support for their applicability over the broad surrounding Rocky Mountain region (IMPROVE sites at Mount Zirkel and Rocky Mountain National Park) are provided in Supplement Text S1 and Figure S7."

3. The figure numbers have been updated due to the inclusion of new figure.

4. A new funding number (DE-SC0021116) that supports this work has been included in the Acknowledgements section.

---

## Author Comment (AC2)

Zhou et al. presented a study on the variability, source apportionment and parameterization of ice nucleating particles (INPs) in the Rocky Mountain during the Surface Atmosphere Integrated Field Laboratory (SAIL) campaign. INP number concentration was measured over nearly two years using an offline droplet freezing assay, which the authors used to analyze the variability of INPs abundance. In addition, positive matrix factorization analysis was performed to investigate the major sources for observed INPs. The contribution of heat-labile and organic materials to the observed INPs was also tested by testing the remained INP abundance of $H_2O_2$ and heat treated samples. The authors also proposed a two-equation parameterization considering the seasonal variability of biological INPs to improve INP prediction. In general, this paper reports significant data on INP abundance and variations in Rocky Mountain and presents important results on INP sources, which are significant to under aerosol-cloud interactions in this region. However, additional details on the measurement and data analysis are expected to be provided in the revised version, along with a more in-depth discussion of the correlations between INPs and aerosols to more effectively present the narrative of this study. We hope that our comments will help the authors improve the revised version of this preprint. We recommend the acceptance for publication in *Atmospheric Chemistry and Physics* (ACP) after appropriate revision.

We thank the reviewer for the positive evaluation of our work and for the thoughtful comments. All comments have been carefully addressed in the revised manuscript.

**General comments:**

**Abstract should be more concise by rephrasing the first three sentence. INP source apportionment should be stressed more. It should be clearly noted that the major/dominant INP sources are locally emitted coarse-sized dust particles and biological particles. This is the key.**

Thank you for this comment. We would like to clarify that the methods used are able to suggest the dominant aerosol types that the INPs are associated with, but cannot directly apportion INPs to sources, as INPs are a small fraction of the total aerosol that is used to characterize the influences of various sources. We make this point clearer in the revised manuscript.

The Abstract were revised to make it more concise as follows:

"Atmospheric ice-nucleating particles (INPs) significantly influence cloud microphysics and aerosol-cloud interactions. Understanding INPs in mountain regions is important for predicting impacts on regional clouds and precipitation." (Lines 13–15)

The discussion of the source of aerosol and their relationship with INPs has been revised to highlight the major sources as follows:

"Aerosol sources were resolved, and INP concentrations were correlated with a coarse dust aerosol type, which dominates $PM_{10}$ in this region. Calculating IN active surface site densities

($n_s$) further supporting the primary contribution from coarse dust to INPs. Treatment with $H_2O_2$ indicated substantial contributions (91% on average) from organic INPs across all activation temperatures, suggesting that supermicron organic-containing soil dust dominates the INPs in this region. Heat-labile INPs, likely biological in origin, were identified as dominant at $> -15\ °C$ through heat treatment of samples and showed significantly lower contributions in winter (~96% reduction)." (Lines 20–24)

**The classification of five PMF factors should be clearly defined in the main text but not in the SI. What is the size difference between Coarse dust and Fine dust? More details about the difference between these two should be provided. Also, specific properties associated with these five factors should be provided. For example, if it is the case, one could state that 'the mass concentration of elemental carbon and K element was used for traces for biomass burning aerosols (BBA)'. Also, the information of instruments/measurements for those properties to represent five factors should be provided.**

The PMF analysis and the classification of PMF factors have been added in the manuscript as follows:

"A five-factor solution was selected as the optimal solution based on the $Q/Q_{exp}$ value and interpretation of the physical meanings of the factors (Brown et al., 2015). The corresponding factor profiles and time series are shown in Figures S5 and S6. These factors were identified, based on chemical signatures and previous literature, as coarse dust, fine dust, biomass burning, sulfate-dominated, and nitrate-dominated sources. Coarse and fine dusts had high contributions from Al, Ca, Fe, Mg, and Si, which are the main components of mineral dust (Liu and Hopke, 2003). Coarse dust explained more than 90% of the coarse mass ($> PM_{2.5}$), while there was no contribution from coarse mass in the fine dust factor. The biomass burning factor was strongly associated with organic and elemental carbon, which are mainly from combustion processes, and K, a tracer of biomass burning (Hopke et al., 2020). The other two factors are dominated by nitrate and sulfate, which are related to the formation of secondary aerosols and possibly some primary emissions from regional sources that include energy production and distant urban regions. Some similar factors were also resolved in published PMF analyses using IMPROVE data (Liu and Hopke, 2003; Hwang and Hopke, 2007)." (Lines 219–232)

Coarse dust explained more than 90% of the coarse mass ($> PM_{2.5}$), while there was no contribution from coarse mass in the fine dust factor. This comparison indicates that fine dust represents particles smaller than 2.5 µm in aerodynamic diameter, while most coarse dust consists mainly of particles larger than this size. This clarification has been added to the manuscript. The characteristics of the PMF factors have also been explained.

The information describing the analytical techniques used for the IMPROVE chemical dataset has been added as follows:

"In the IMPROVE program, elemental analysis was performed on the Teflon filters using X-ray fluorescence (XRF), water-soluble anions were analyzed by ion chromatography (IC), and elemental carbon (EC) and organic carbon (OC) were analyzed using a thermal-optical carbon analyzer (Hand, 2023)." (Lines 210–213)

"Hand, J.: IMPROVE Data User Guide 2023 (Version 2), Interagency Monitoring of Protected Visual Environments (IMPROVE) Program, available at: https://vista.cira.colostate.edu/Improve/data-user-guide/, 2023."

**BBA containing elemental carbon or black carbon particles are reported to be poor INPs in the mixed-phase cloud regime (Gao et al., 2025; Wieder et al., 2022). It should be BBA associated (co-emitted) soil dust or organic particles that contribute to observed INPs (Mccluskey et al., 2014). This is also supported by the $n_s$ results in Figure 4b where it shows the $n_s$ of BBA period samples is similar to those samples with dust as INP sources. This point should be delivered to the readers more clearly. Instead, it now reads like the authors classified BBA as one of the INP sources which seems all emitted particles contribute to the observed INPs. In addition, would it be possible if the authors can also provide fire maps (e.g., from NASA FIRMS) with HYSPLIT airmass back trajectories for INP samples influenced by BAA? This will provide further evidence of BAA.**

During biomass burning events, aerosol number concentrations were significantly enhanced, especially for submicron particles (Figure 2). However, the $n_s$ of INP samples for which the ambient aerosols were dominated by biomass burning were similar to those of coarse dust, and were higher than $n_s$ values reported from laboratory biomass burning studies (Umo et al., 2015; Jahn et al., 2020) and ambient biomass burning observations (McCluskey et al., 2014; Barry et al., 2021b; Zhao et al., 2024). This may indicate that coarse dust still made a primary contribution to INPs in these samples, as coarse dust has a much higher $n_s$ than those from biomass burning. To make this point more clearly, the sentences have been revised as follows:

"However, its contributions to INPs could be affected by coarse dust. After normalization by surface area ($S_{m,500}$), the $n_s$ of INP samples for aerosols dominated by biomass burning were similar to those of coarse dust-dominated samples. Compared with previous studies (comparison based on computing $n_s$ using total surface area), these values were higher than $n_s$ reported from laboratory biomass burning studies (Umo et al., 2015; Jahn et al., 2020) and those reported in ambient biomass burning observations (McCluskey et al., 2014; Barry et al., 2021b; Zhao et al., 2024). During SAIL, this finding may have been due to the presence of coarse dust, which has a much higher $n_s$ than biomass burning, as this aerosol type still contributed moderately to the total aerosols in these samples (an average of 22% of $PM_{10}$), although biomass burning was the dominant aerosol source (62% on average). Wildfire events could also be a source of airborne dust (Wagner et al., 2018; Meng et al., 2025)." (Lines 474–483)

Furthermore, the terms "INPs (or $n_s$) related to coarse dust/biomass burning/fine dust" used in the discussion section might be misleading, as the dominant aerosol source being biomass burning does not imply that the INPs necessarily originate all from biomass burning. To clarify, terms have been revised as follows:

"Compared to INPs samples that aerosols dominated by coarse dust, fine dust-dominated time periods showed lower INP concentrations (Figure 5)." (Lines 452–453)

"Besides their different origins, the higher INP concentrations associated with coarse dust-dominated samples compared to those dominated by fine dust can also be attributed to differences in particle size." (Lines 457–459)

"INPs related to biomass burning dominated samples presented comparably high concentrations, which may be related to the significantly elevated aerosol loading during biomass burning events." (Lines 460–462)

"After normalizing by surface area, the $n_s$ for the fine dust-dominated samples showed closer values with those of the coarse dust-dominated samples, while still lower. This suggests that the lower INP concentrations in fine dust-dominated samples can be partly attributed to differences in aerosol surface area concentrations, but also to lower active site density due to potentially different INP sources. The differences in $n_s$ between coarse dust- and fine dust-dominated samples were limited, likely because there were still small contributions from coarse dust (17% on average), although fine dust dominated these samples (59% on average)." (Lines 477–484)

**It is good that the authors provided results on DNA analysis for Snomax influenced samples in the Supplementary. Why not also for heated and H2O2 treated and untreated samples? This will provide direct and strong evidence on the presence/contribution of biological particles to the observed INPs. In Section 3.5, the results of heated and H2O2 treated samples are in-direct evidence on the contribution of biological particles to INPs.**

For Snomax-influenced samples, we further performed DNA analysis, partly because *P. syringae* is the known target in Snomax and can be used to identify the impact of Snomax. However, there is no prior information on the specific types of biological INPs in this region. It may be possible to identify certain INP-active bioaerosols using biological analytical methods, but such work requires extensive analyses and falls outside the scope of this study, which focuses on the seasonality, major sources, and parameterization of INPs in the Rocky Mountains. Therefore, this topic is better addressed in a separate investigation.

For the heated and $H_2O_2$-treated samples, as there is no prior information on the specific INP-active bioaerosols in this region, DNA analysis would not provide meaningful insight into the biological INPs.

Currently, based on our methodology, we can only provide indirect evidence of the biological INPs, or "biological/heat-labile INPs", for the type of INPs removed and remaining after heat treatment. Given their warm activation temperature ranges and considering concurrent bioaerosol measurements during the SAIL campaign, these biological/heat-labile INPs are presumably biological INPs, although further identification (as bacteria, fungal spores, pollen, etc. remains a topic for future development.

**Some statements in Section 4 are repeating discussions in Section 3, which makes it unnecessary long and reads not very interesting. Also, section 4 should be divided into two sections, one for summary/conclusion and one for atmospheric implications. In the original manuscript, atmospheric implications are not well/sufficiently addressed as it appears in the section title.**

To address the redundant statements and discussions that had already been covered in Section 3, these unnecessary discussions have been removed from Section 4. To improve the clarity and readability of the paper structure, Section 4 has been reorganized: instead of separating the content into two short sections, which is not essential given the overall length, the revised Section 4 now begins with a concise summary/conclusion paragraph followed by the atmospheric implications of this study.

In addition to the atmospheric implications already discussed in the manuscript, we have further highlighted the following points in the revision:

1. This study identified the major INP source by linking long-term INP measurements with aerosol source apportionment, presenting reasonable results that agree well with other studies (DeMott et al., 2025; Lacher et al., 2025). This approach may have broader applicability for INP source attribution in other regions. Future studies to provide direct chemical information on INP would be useful.
2. We also expanded the discussion on the possible sources of biological INPs, such as bacteria and pollen, which refers to previous studies of bacteria and pollen in the Rocky Mountain region.

Section 4 has been rewritten as follows:

[revised manuscript text omitted]

**Specific comments:**

**Line 27 & 30: Provide number/statistics for 'Substantial' and 'significant'. This makes the abstract stronger.**

Statistical values have been added to the Abstract to support the findings as follows:

"Treatment with $H_2O_2$ indicated substantial contributions (91% on average) from organic INPs across all activation temperatures, suggesting that supermicron organic-containing soil dust dominates the INPs in this region. Heat-labile INPs, likely biological in origin, were identified as dominant at > −15 °C through heat treatment of samples and showed significantly lower contributions in winter (~96% reduction)." (Lines 22–26)

Statistical values have also been added in the discussion as follows:

"This separation clearly shows that on average, INP concentrations were lower in cold seasons, with the most striking difference at temperatures warmer than −15 °C (15% of INPs in other seasons)." (Lines 505–507)

"The difference between the base and heat spectra indicated a large contribution (82–94%, 90% on average) of heat-labile INPs at warm temperatures (> −15 °C), which are presumably biological INPs." (Lines 518–520)

"Biological/heat-labile INPs during the cold seasons account for only 4% of those in the other seasons." (Lines 533–534)

**Line 51-52: colder than which temperature?**

This sentence has been revised as follows:

"Atmospheric mineral dust particles are considered a dominant contributor of INPs throughout much of the troposphere (Murray et al., 2012; Hoose and Möhler, 2012), and they produce high INP concentrations in a mass or surface area basis at temperatures lower than −15 °C (Atkinson et al., 2013; Kiselev et al., 2017)." (Lines 46–49)

**Line 89-101: The paragraph develops general statements so it should be moved forward**

Paragraph in lines 89-101 introduced the mountain regions and the importance of studying INPs in these regions. The previous paragraphs focus on INPs, their sources, and INP parameterizations. This paragraph transitions the introduction toward mountain systems and provides a logical bridge to the subsequent description of the study area. Therefore, we believe that keeping it in its current position is more appropriate than moving it earlier, where it would become mixed with the more general introduction of INPs.

**Line 154: reference to Vali 1971 does not follow ACP guidelines. Alos correct others, like Line 250**

The citations have been corrected as follows:

"Cumulative INP concentrations as a function of temperature ($n_{\text{INPs}}$(T), INPs per liter of air) were calculated based on the method of Vali (1971) using:" (Lines 149–150)

"To account for the peak in occurrence near the sampling site, the residence times were further normalized by the distance from the SAIL sampling site, following the method of Ashbaugh et al. (1985)." (Lines 267–269)

**Line 155-156: not clear about the field blank collection before removal and storage. Remove what? Do you have filed blank for every sample for background noise correction? Also details about INP data processing is missing. Simply referring to Vali 1971 is not enough. There are differences in calculating the INP numbers.**

The calculation equation of INP concentration has been added as follows:

"Cumulative INP concentrations as a function of temperature ($n_{\text{INPs}}$(T), INPs per liter of air) were calculated based on the Vali method (Vali, 1971) using:

$$n_{INPs}(T) = ln(\frac{N_0}{N_0 - N(T)}) \times \frac{V_w}{V_c} \times \frac{1}{V_a}$$

where $N_0$ is the total number of wells containing aliquots, $N(T)$ is the cumulative number of wells frozen at temperature $T$, $V_w$ is the volume of water used for particle resuspension, $V_c$ is the aliquot volume added to each well, and $V_a$ is the total sampled air volume." (Lines 148–153)

The frequency of blank filter collection and the collection method have been revised as follows:

"Field blank filters were collected every month by briefly exposing them at the sampling site for several seconds before storage." (Lines 154–157)

**Line 167: how much of H2O2 was added? And samples should be termed H2O2-heat treated but not only H2O2 treated. This should be changed through the paper**

30% $H_2O_2$ was added to the solution to make a final mixture concentration of 10%. The sentence has been revised as follows:

"In the peroxide treatment, 30% $H_2O_2$ was added to the solution to make a final concentration of 10%, and the mixture was heated at 95 °C for 21 min under UVB light to digest organics (Suski et al., 2018)." (Lines 166–169)

The heating process here is part of the $H_2O_2$ digestion process, together with UVB illumination, to generate hydroxyl radicals for organic removal. Also, all samples subjected to $H_2O_2$ treatment have been heat-treated, which removes heat-labile INPs first. For clarity, we use the terms "$H_2O_2$ treatment" and "heat treatment" consistently throughout the paper. This method and terminology have also been introduced in previous studies (Suski et al., 2018; Testa et al., 2021)

Suski, K. J., Hill, T. C., Levin, E. J., Miller, A., DeMott, P. J., and Kreidenweis, S. M.: Agricultural harvesting emissions of ice-nucleating particles, Atmos. Chem. Phys., 18, 13755-13771, 2018.

Testa, B., Hill, T. C. J., Marsden, N. A., Barry, K. R., Hume, C. C., Bian, Q., Uetake, J., Hare, H., Perkins, R. J., Möhler, O., Kreidenweis, S. M., and DeMott, P. J.: Ice Nucleating Particle Connections to Regional Argentinian Land Surface Emissions and Weather During the Cloud, Aerosol, and Complex Terrain Interactions Experiment, J. Geophys. Res: Atmos., 126, https://doi.org/10.1029/2021jd035186, 2021.

**Line 187-190: How long can it affect the campaign sampling? It may also influence subsequent sampling but not only overlapped samples? can you evaluate this? since you see the Snomax effects**

The cloud seeding activities near the SAIL campaign site are conducted in less than 24 hours, typically for 4-8 hours. Aerosol samples were collected every three days. Only samples whose

sampling periods overlapped with cloud seeding activities were considered affected. This is supported by the clear contrast in the INP temperature spectra between seeding-affected samples and other winter samples. Therefore, cloud seeding activities would only influence the overlapping samples and are unlikely to impact subsequent samples collected 3 days later.

This point has been added in the revision as follows:

"Cloud seeding activities last less than 24 hours, typically 4-8 hours, and are unlikely to affect the subsequent sample collected 3 days later." (Lines 190–192)

**Line 214: provide size ranges for Coarse dust and Fine dust? in SI text S1 you provided that they contain similar elements Al, Ca, Fe, Mg, and Si but what is their difference making it Coarse or Fine?**

Coarse dust and fine dust factors both contain elements of Al, Ca, Fe, Mg, and Si. However, the coarse dust factor explained more than 90% of the coarse mass concentrations, calculated from $PM_{10}$−$PM_{2.5}$ mass concentrations, while there is no contribution from that variable in the fine dust factor. This main difference distinguishes the coarse dust and fine dust factors. This point has been clarified in the main text as follows:

"Coarse and fine dusts had high contributions from Al, Ca, Fe, Mg, and Si, which are the main components of mineral dust. Coarse dust explained more than 90% of the coarse mass ($> PM_{2.5}$), while there was no contribution from coarse mass in the fine dust factor." (Lines 225–227)

**Line 273: should be ice nucleation site according to Vali et al. (2015)**

The terminology has been revised as follows:

"Assuming that the number of active ice nucleation sites is linearly proportional to the particle surface area." (Line 292)

**Line 280-281: Did the authors test the role of particles smaller than 500 nm? Gao et al. (2024) reported that particles smaller than 500 nm may contribute to INPs active in the mixed-phase cloud regime by comparing the correlations between INPs at -25C and SMPS+APS particles or APS particles.**

In Gao et al. (2025), they found that for samples collected in the free troposphere, INP concentrations were significantly positively correlated with particles larger than 500 nm and had no significant correlations with particles smaller than 500nm. For the sampling within the PBL, they found that particles smaller than 1 μm were significantly correlated with INPs, whereas

particles larger than 1 µm were weakly linked to INPs. They suggested this correlation may be due to fine particles having much higher number concentrations than coarse mode particles, and also because particles from various sources that span different size ranges are responsible for the observed INPs in the PBL. Furthermore, when they considered the sampling days with and without dust events, both showed that only particles in size ranges larger than 500 nm had significant correlations with INPs.

In this study, we found that the coarse dust factor correlated well with INPs at multiple activation temperatures, whereas correlations with the fine dust factor were insignificant (Table 1 in the manuscript). These kinds of correlations are similar to what Gao et al. (2025) reported (correlation between aerosol sizes and INP concentrations), although with a different threshold (around 2.5 µm for coarse dust and fine dust). These correlations provide indirect insight that smaller particles may contribute less to INPs in this study.

However, aerosol size distributions are largely dominated by non-ice-active particles. To understand the size distribution of INPs, size-resolved sampling is needed. In a recent preprint, Lacher et al. (2025) directly measured the sizes of INPs and found that supermicrometer particles contributed significantly to INPs in the Rocky Mountain region, which is similar to our study area. This point has been added as follows:

"Coarse dust showed good correlations with INPs active at all temperatures (Table 1), with correlation coefficients increasing for colder temperatures, suggesting that coarse dust is a major source of INPs, particularly at lower temperatures. This is consistent with previous findings that dust dominates the INPs at temperatures below −20 °C (Beall et al., 2022; Testa et al., 2021; Kanji et al., 2017). Furthermore, Lacher et al. (2025) provided direct evidence that INPs active at cold temperatures were significantly contributed by supermicrometer particles, which they attributed to dust, in the Rocky Mountains." (Lines 385–392)

**Lien 285: maybe move S15 as S9? the previous reference is S8 in line 252**

Figure S15 has moved to Figure S10.

**Line 296-300: prepare a figure and provide it in the SI? Showing/comparing INP abundance in previous studies, like Lacher et al. (2025) and Tobo et al. (2013)**

In Lines 296–300, we provided INP concentration values at −15 °C and −25 °C for readers to understand the INP levels in the Rocky Mountain region and to more easily compare with other studies. In the following sentence, we compared our values with those summarized by Kanji et al. (2017). Although it would be helpful to present such a comparison in a figure, it is difficult to obtain exact INP concentration values from some studies, for example, Tobo et al. (2013) and Tobo et al. (2014) only showed their results in figures without providing numerical values, making is difficult to determine precise concentrations.

In addition to comparing the INP concentrations, comparing the parameterization lines based on the IN active surface site density derived from their measurement is also meaningful. This comparison has been provided in Figures S10 and S14.

Here, we have now included the comparison with Lacher et al. (2025) as follows:

"INP concentrations ranged from $4 \times 10^{-4}$ $L^{-1}$ to 1.5 $L^{-1}$ (mean: 0.15 $L^{-1}$, median: 0.05 $L^{-1}$) at $-15$ °C, and from 1.2 $L^{-1}$ to 90 $L^{-1}$ (mean: 16 $L^{-1}$, median: 12 $L^{-1}$) at $-25$ °C. This is comparable to online INP measurements in the Rocky Mountain region (median: 8.2 $L^{-1}$ at $-26$ °C; Lacher et al., 2025)." (Lines 313–316)

**Line 304-305: isn't it also the case for T-20 T-15 showing peaks in Sep 2021? and also T-10?**

Based on Figure 3a, for active temperatures at $-10$ °C, $-15$ °C, and $-20$ °C, the highest INP concentrations were observed in May or June. The monthly mean INP concentrations shown in Figure 3b also present this trend.

**Line 308-309: An increase in total aerosols does not necessarily mean an increase in INPs. Like not all BBA particles like BC contribute to INPs at -25C. Please give in-depth insight.**

The original sentence was ambiguous and could be interpreted differently from what we intended. We intended to say that intense wildfires increased aerosol loading, and that the smoke may also have contributed to the observed enhancement in INP concentrations in September 2021. To clarify, this sentence has been revised as follows:

"Intense wildfires occurred in that region during the summer of 2021 (Jain et al., 2024), and the transported smoke plumes increased aerosol loading at the SAIL site. These smoke intrusions may also result in enhanced INP concentrations active at low temperatures." (Lines 327–330)

**Line 320: why emission decrease? Like because of snow cover and decreased metabolism**

The possible reasons for lower bioaerosols in winter are added as follows:

"Bioaerosols are typically recognized as major INP sources at these activation temperatures (Kanji et al., 2017), and their emissions generally decrease in winter in most areas due to reduced biological activities and snow cover limiting resuspension (Fröhlich-Nowoisky et al., 2016)." (Lines 340–342)

**Line 323-324: evidence/relevant results on dust events?**

From the PMF results, coarse dust showed the highest concentrations in June 2022. Therefore, the possibility of dust contribution to INPs even at warm activation temperatures cannot be ruled out, and we claimed that "which may be related to a specific biological emission event and/or dust event."

**Line 328: be more specific. Dust is more relevant INPs for T<-15C**

The sentence has been revised as follows:

"The elevated INPs from April to September may be attributed to enhanced dust aerosols, as dust concentrations were found to increase during this period (Hand et al., 2017), and dust is a significant source of INPs, especially at temperatures below −15 °C (Kanji et al., 2017; DeMott et al., 2015)." (Lines 352–355)

**Line 329: Does lower-temperature INPs refer to INPs at -20 and -25C?**

Yes, this is the discussion of INPs at activation temperatures of −20 °C and −25 °C, as indicated at line 341. This sentence has been revised to be more specific, as explained in the last comment.

**Line 335: Section 3.2: Figure S6 for SAIL campaign should be moved to this section. Also consider that this section should be harmonized with Section 3.1 discussing also in a seasonal manner**

Figure S6 showed the time series of PMF factors. Section 3.2 discussed the relationship between aerosol sources and INPs. Figure S9, which showed both INPs and source concentrations, is more relevant to this topic and has been added to the main text as Figure 4.

The seasonality has been discussed in section 3.1. Section 3.2 focused on the relationship between aerosol sources and INPs. Therefore, we think it is better to discuss seasonality when discussing air mass source types.

**Line 345-347: I am disappointed that a statement is missing. figure S6 BBA peak in Sep 2021 is a very strong evidence for INP peak in Figure 3b. A clear statement should be made**

Lines 345–347, located in the first paragraph in section 3.2, showed the results of the PMF analysis, including the variations of each aerosol source during the SAIL campaign, to present a

general view of the variations in aerosol influences at the site. In this paragraph, the relationship between aerosol sources and INPs is not included, which is discussed in the following two paragraphs.

The biomass burning aerosol peak in September 2021 indeed coincided with the high INPs (−25 °C) peak, suggesting an important contribution from biomass burning aerosol. This can be more clearly identified from Figure S9 (now moved to the main text as Figure 4), showing the relationship during the whole campaign period. These points have been discussed in lines 382–385 and 398–403. We note that part of the peak in INPs in this time period can be attributed to higher aerosol loadings. Once the INP spectra were normalized by surface area (Figure 5b), the biomass-burning-dominated sample became consistent with the others. Combined with comparisons of the SAIL $n_s$ for this air mass type, with those reported in the literature for lab studies specifically on biomass burning samples (ref from Reviewer 1), showing the SAIL values to be much larger, it may be that coarse dust associated with the biomass burning-dominated air masses was the actual contributor to elevated INPs. We now discuss this in more detail, pointing out the correlation as the Reviewer suggests, but also the caveats noted here.

"During biomass burning events, aerosol number concentrations were significantly enhanced, especially for submicron particles (Figure 2). A correlation was found between the biomass burning factor mass concentrations and the total surface area concentrations of aerosols (Figure S12), suggesting that such events significantly increased aerosol surface area concentrations. However, its contributions to INPs could be affected by coarse dust. After normalization by surface area ($S_{m,500}$), the $n_s$ of INP samples for aerosols dominated by biomass burning were similar to those of coarse dust-dominated samples. Compared with previous studies (comparison based on computing $n_s$ using total surface area), these values were higher than $n_s$ reported from laboratory biomass burning studies (Umo et al., 2015; Jahn et al., 2020) and those reported in ambient biomass burning observations (McCluskey et al., 2014; Barry et al., 2021b; Zhao et al., 2024). During SAIL, this finding may have been due to the presence of coarse dust, which has a much higher $n_s$ than biomass burning, as this aerosol type still contributed moderately to the total aerosols in these samples (an average of 22% of $PM_{10}$), although biomass burning was the dominant aerosol source (62% on average). Wildfire events could also be a source of airborne dust (Wagner et al., 2018; Meng et al., 2025)." (Lines 471–485)

**Line 364-366: We cannot agree with this statement. Coarse dust include dust particles and dust carrying biological materials. Dust is not effective INPs at -10C, but BIO-dust could be. Also not all coarse particles carrying biological materials and behave as active INPs. Also, airborne bacteria can be submicron as part of biological particles.**

In lines 364–366, the sentence showed the interesting correlation between coarse dust and INPs at −10 °C ($R^2 = 0.43$), a temperature range usually associated with biological INPs. The possible reasons for this correlation were explained in the next sentence: "This may be due to the large number of coarse dust particles, biological INPs carried on dust particles, and/or the inclusion of biological particles in the coarse dust factor, as biological particles are mostly supermicron in

size (Després et al., 2012)." This sentence already includes the possibilities given by the reviewer.

**Line 392: how can you see it is the predominant source by referring to Figure S6? does S6 provide fraction or other results to support this argument?**

To better illustrate how dominant sources were assigned, a new figure showing the proportions of each aerosol type contribution to the aerosols has been added to the SI.

[Figure]

**Figure S8.** Proportions of source contributions for all samples. The label below each sample indicates its classification: CD (coarse dust), B (biomass burning), D(dust), FD (Fine dust), M (Mixed sources), and N (no source information).

**Line 392-394: tune down. Statement is too strong, given that in Sep 2021 BBA associated aerosol is the controlling source**

Although the biomass burning aerosol showed a distinct high peak in September 2021, coarse dust also showed elevated concentration in this month. Furthermore, as explained in the response above, once the INP spectra were normalized by surface area (Figure 12), the biomass-burning-dominated sample became consistent with the others. During SAIL, this finding may have been due to the presence of coarse dust, which has a much higher $n_s$ than biomass burning, as this aerosol type still contributed moderately to the total aerosols in these samples (an average of 22% of $PM_{10}$). Therefore, based on the peak in September 2021 of biomass burning, we cannot

conclude that INPs in those samples are dominated by biomass burning. Coarse dust may also play an important role in INPs in those samples.

However, we agree that the original sentence was too conclusive and has been revised as follows:

"Considering that coarse dust also showed a strong correlation with INPs, this suggests INP concentrations were likely primarily influenced by coarse dust in this area." (Lines 424–426)

**Line 394-395: does it mean the active IN of mineral and soil dust is because of organics and/or salts? I am afraid of that one will not agree. Also, what is point for putting many references? Suggest referring to a literature with clear rationales**

In this sentence, the clause "that contains abundant organics and/or salts" attaches only to "soil dust" and not to "mineral dust". So, this sentence is introducing multiple studies that have investigated the INP sources of mineral dust (DeMott et al., 2003; Niemand et al., 2012; Atkinson et al., 2013; DeMott et al., 2015), soil dust that contains abundant organics (Tobo et al., 2014; Steinke et al., 2016; O'Sullivan et al., 2014; Testa et al., 2021; Pereira et al., 2022), and soil dust that contains salts, name playa dusts (Pratt et al., 2010; Hamzehpour et al., 2022). To clarify, three or less references were cited for each type of INP source, and the sentence has been revised as follows:

"Mineral dust (DeMott et al., 2003; Niemand et al., 2012; Atkinson et al., 2013), soil dust that contains abundant organics (Tobo et al., 2014; Steinke et al., 2016; O'Sullivan et al., 2014), and playa dusts (Pratt et al., 2010; Hamzehpour et al., 2022) have been widely investigated, and are considered as important INP sources." (Lines 426–429)

**Line 416-417: Again, high aerosol number concentration does not necessarily mean more INPs. It is probably because of soil dust and some organics in BBA plumes**

We agree with the reviewer's assessment, as noted above. This sentence has been revised as follows to consider other possibilities:

"INPs related to biomass burning-dominated samples presented comparably high concentrations, which may be related to the significantly elevated aerosol loading during biomass burning events." (Lines 447–449)

As the reviewer suggested, high aerosol loading does not necessarily mean more INPs, it is also related to IN active surface site density and possibly contribution from soil dust or other sources. These points have been discussed in the original manuscript in the next section in lines 471–482.

**Line 431-432: Don't agree. This may suggest coarse dust is more active INPs. Major contributor means more abundant INP number concentrations**

This sentence has been revised as follows:

"The $n_s$ of samples dominated by coarse dust was similar to or slightly higher than those having both abundant coarse and fine dust (categorized as dust), suggesting that INPs from coarse dust have higher IN active surface site density." (Lines 461–464)

**Line 440-441: What is the biomass burning mass concentration IN Figure S11? what does it contain? We only see a statement in Text S1: The biomass burning factor was strongly associated with organic and elemental carbon, which are mainly from combustion processes, and K, a tracer of biomass burning.**

In Figure S11, the concentration of biomass burning was shown in blue dot line and scaled by the right y-axis.

The biomass burning factor profile is shown in Figure S5a.

A new figure (Figure S8) has been added to show the proportion of source contributions to each sample as explained above.

**Line 464: why -18C? not -15C or -20C? It is not very clear for us to read the so-called 'clear segregation'. Instead of cumulative INP and ns curves, we believe differentiate curves will help the authors present more pronounced results.**

The onsets of the segregation into two groups occurred over a temperature range (−17 °C to −19 °C) for two-year samples, rather than at clear single point (like −18 °C). This is because individual samples could show slightly different transition points. For the purpose of comparing warm temperature INPs between samples from cold and other seasons, identifying the exact onset temperature is not essential. The expression of "clear segregation" was misleading and has been removed in the revision. The sentence has been revised as follows:

"From the INP temperature spectra and $n_s$ for all samples (Figure 5), the spectra for INPs active at temperatures higher around −18 °C showed a segregation into two groups: one with higher INP concentrations (and $n_s$) and measured detectable freezing onset temperatures mostly > −10 °C, and the other with lower INP concentrations (and $n_s$) and lower measured detectable freezing onset temperatures, with most of those samples assigned to aerosol sources of mixed and fine dust." (Lines 486-488)

The segregation temperature also relates to the two-equation parameterization developed in this study, in which the two equations were constrained to intersect at −20 °C. Because the two

equations are very similar between −20 °C to −18 °C (Figure 9), the exact choice of the intersect point is not critical in practice and may be considered somewhat arbitrary. Also, estimation based on these two equations reproduces the measurements well. As shown in Figure 6, the $n_s$ values from different periods clearly converge to a similar range at temperatures colder than −20 °C.

As the reviewer suggested, we plot the differential spectra (k(T)) based on Vali (1971) and Vali (2019) for our measurements.

[Figure]

Panel a and b show the differential INP spectra for samples from other seasons and cold seasons, respectively. As most samples showed nearly log-linear INP concentration spectra (Figure 5 in the manuscript), the differential spectra also showed a similar log-linear behavior. There were clear differences between samples collected during cold versus other seasons, with cold-season samples showing lower differential nucleus concentration k(T) at warm temperatures. However,

the transition temperature, which occurred around −18 °C (−17 °C to −19 °C), is still not clear in this figure. Further zoom in and compared samples from two periods (panel c, samples from other seasons in blue and from cold seasons in red) showed there are differences in k(T) for temperatures warmer than a transition temperature (around−18 °C but not clear for a specific point). This is partly due to systematic uncertainties associated with our droplet-freezing INP measurement method, which counts frozen wells. These uncertainties introduce noise to the differential curve, and the changes in k(T) caused by different INP types may be obscured when the changes are not strong. This can be better seen in the single sample differential curve (panel d). The k(T) could be more useful for the case of the Snomax-affected sample (panel d), which has a significant change.

Therefore, the differential spectra are not included in the manuscript.

**Line 469-470: In Kanji et al. (2017), the temperature is -15C but not -18C**

The sentence has been revised as follows:

"At freezing temperatures warmer than −15 °C, biological INPs are likely to play a more important role (Kanji et al., 2017)." (Lines 493-494)

**Line 480: again, we hardly read this -18C in Figure 5**

As explained above, the transition point is around −18 °C rather than a clear point, all related sentences have been revised as follows:

"A further comparison of $n_s$ (Figure 6b) showed that samples from cold seasons had similar $n_s$ at temperatures colder than approximately −18 °C." (Lines 508–509)

"These results suggest that INPs that activated at temperatures colder than around −18 °C likely originated from similar sources throughout the whole year, which were primarily associated with coarse dust, as discussed above." (Lines 510–513)

"However, in cold seasons, the contribution from biological INPs was significantly reduced, leading to the divergence in the spectra for temperatures warmer than around −18 ˚C." (Lines 513–515)

**Line 510: what is the correlation coefficient? and p-value? is it significant?**

The correlation coefficients were provided in Figure 8. The p-value of each correlation has been added in Figure 8, and a marker for each panel has been labeled. The revised Figure 8 is shown here.

[Figure]

**Figure 8.** Correlations between concentrations of (a, b) biological/heat-labile INPs and other organic INPs, (c, d) biological/heat-labile INPs and inorganic INPs, and (e, f) other organic INPs and inorganic INPs, active at either −15 °C (upper row) or −25 °C (lower row). Dashed lines indicate a 10:1 relationship for reference, and the orange rectangles highlight the strongest correlations at each temperature. (Lines 765–770)

**Line 530: please refer to subpanels. Also for other statements for the discussion on results in Figure 7.**

The subpanels have been added in Figure 7 (Figure 8 in the revision) and referred to in the discussion.

**Line 576-577: change the wording 'single'. This may cause misunderstanding because you have two equations in the parameterization**

To avoid possible misunderstanding of this sentence with the next one, this sentence has been revised as follows:

"The agreement between predicted and measured INPs based on these two equations showed improvement compared to using the one equation method above (Figure S15), better representing the measured INPs across the full measured temperature range (Figure 9)." (Lines 610–613)

**Line 590: Is there any result/evidence provided for showing intensive biological INP events?**

This sentence has been removed to make a more concise summary section as explained in the response to the fifth general comment.

Here is the answer to the reviewer's question. The sentence in line 590 can be referred to Figure 3b, which shows the monthly mean INP concentrations. At temperatures of −10 °C and −15 °C, which are typically recognized as temperatures contributed by biological INPs, distinct peaks were observed in June 2022. These peaks were much lower in June 2023, suggesting there was a biological INP emission event in June 2022.

The intensive biological INP event in June 2022 can also be identified in heat treatment spectra shown in Figure S1. The samples from June 2022 showed elevated INP concentrations at warm temperatures ($> −15$ °C) that were significantly decreased after heat treatment.

The abundance of fluorescent bioaerosol particles in June 2022 was also investigated using a Wideband Integrated Bioaerosol Sensor by Shawon et al. (2025).

**Line 609-610: Is there any result presented for the correlations between fine dust and INPs?**

The correlation coefficients and p-value of the correlation between fine dust and INPs are shown in Table 1.

Figure S9 has been moved to the main text as Figure 4 to show the relationships between air mass source factors, including fine dust, and INP concentrations. We again caution that the source apportionment is for total aerosols and not specifically for INPs. We have tried to make this clearer throughout the text.

**Technical corrections:**

**Figure 3a colour is difficult to read. Suggest also to use different symbols or find other better ways to visualize the data**

To improve the readability of Figure 3a, the colors, markers, and size of the figure have been modified as follows:

[Figure]

**Figure 4a may use shedding ranges to better present the data**

We assume that the reviewer is referring to "shading ranges". Plotting shading ranges of samples dominated by different aerosol types would not substantially improve data interpretation and would obscure information from individual samples; So, shading ranges are not included.

**Figure 7: what is the significance level of the calculated coefficents?**

The significance levels (p-value, two-tailed) have been added in Figure 7 (Figure 8 in the revision) as explained above.

Other revisions

1. Citations including "Mccluskey" corrected as "McCluskey".

2. Sentence in line 242 has been revised as follows:

   "Further details on the PMF analysis and results, as well as support for their applicability over the broad surrounding Rocky Mountain region (IMPROVE sites at Mount Zirkel and Rocky Mountain National Park) are provided in Supplement Text S1 and Figure S7."

3. The figure numbers have been updated due to the inclusion of new figure.

4. A new funding number (DE-SC0021116) that supports this work has been included in the Acknowledgements section.